# Dissociation of task engagement and arousal effects in auditory cortex and midbrain

**Daniela Saderi[1,2], Zachary P Schwartz[1,2], Charles R Heller[1,2], Jacob R Pennington[3], Stephen V David[1]\***

[1]Oregon Hearing Research Center, Oregon Health and Science University, Portland, United States; [2]Neuroscience Graduate Program, Oregon Health and Science University, Portland, United States; [3]Department of Mathematics and Statistics, Washington State University, Vancouver, United States

**Abstract** Both generalized arousal and engagement in a specific task influence sensory neural processing. To isolate effects of these state variables in the auditory system, we recorded single-unit activity from primary auditory cortex (A1) and inferior colliculus (IC) of ferrets during a tone detection task, while monitoring arousal via changes in pupil size. We used a generalized linear model to assess the influence of task engagement and pupil size on sound-evoked activity. In both areas, these two variables affected independent neural populations. Pupil size effects were more prominent in IC, while pupil and task engagement effects were equally likely in A1. Task engagement was correlated with larger pupil; thus, some apparent effects of task engagement should in fact be attributed to fluctuations in pupil size. These results indicate a hierarchy of auditory processing, where generalized arousal enhances activity in midbrain, and effects specific to task engagement become more prominent in cortex.

**\*For correspondence:**
davids@ohsu.edu

**Competing interests:** The authors declare that no competing interests exist.

## Introduction

Hearing is a dynamic process that requires integration of the sensory evidence provided by physical attributes of sound with information about the behavioral context in which auditory perception occurs (*Bizley and Cohen, 2013*; *Fritz et al., 2007*). As sound information passes through the brain, coordinated activity in auditory areas extracts information necessary to guide behavior. Compared to passive listening, engaging in a task that requires auditory discrimination leads to changes in neural excitability, as well as in spectral and spatial selectivity (*Downer et al., 2015*; *Fritz et al., 2005*; *Fritz et al., 2003*; *Knudsen and Gentner, 2013*; *Kuchibhotla et al., 2017*; *Lee and Middlebrooks, 2011*; *Otazu et al., 2009*; *Ryan and Miller, 1977*; *Yin et al., 2014*). These changes are often attributed to the specific task demands, enhancing the representation of important sound features by selectively increasing or decreasing the activity across neural populations (*Bagur et al., 2018*; *David et al., 2012*; *Kuchibhotla et al., 2017*). In some studies, changes have been reported to persist for several minutes after the active behavior, while in others, activity and tuning changes were observed to rapidly return to baseline (*Fritz et al., 2003*; *Slee and David, 2015*).

The substantial variability of effects across studies suggests that task engagement activates a constellation of non-auditory signals that comprise a complex internal state. Task-related changes in the activity of auditory neurons have been attributed to temporal expectation (*Jaramillo and Zador, 2011*), reward associations (*Beaton and Miller, 1975*; *David et al., 2012*), self-generated sound (*Eliades and Wang, 2008*), and non-sound related variables, such as motor planning (*Bizley et al., 2013*; *Huang et al., 2019*), motor activity (*Schneider et al., 2014*; *Zhou et al., 2014*), degree of engagement (*Carcea et al., 2017*; *Knyazeva et al., 2020*), and behavioral choice (*Tsunada et al.,*

*2016*). Even in the absence of explicit behavior, fluctuations in neural activity are observed throughout the forebrain, including primary sensory regions (*Ringach, 2009*; *Stringer et al., 2019*). This activity is related to cognitive states such as arousal and may also modulate sensitivity to sensory inputs (*Fu et al., 2014*; *Wimmer et al., 2015*). Therefore, attempting to interpret changes in neuronal activity solely in light of a single binary variable, such as being engaged in a task or not, will lead to an incomplete and even incorrect understanding of how sensory information is processed.

A straightforward way to monitor some task-independent changes in behavioral state is by measuring pupil size (*Kahneman and Beatty, 1966*; *McGinley et al., 2015*; *Zekveld et al., 2018*). In humans, luminance independence changes in pupil size have been shown to correlate with mental effort (*Beatty, 1982*; *Winn et al., 2015*), changes in states of arousal (*Kahneman and Beatty, 1966*), aspects of decision-making (*Gilzenrat et al., 2010*), and task performance (*Schriver et al., 2018*). In animals, fluctuations in pupil size closely track locomotion and neural activity throughout the mouse forebrain (*McGinley et al., 2015*; *Stringer et al., 2019*). Pupil size also tracks the degree of neural synchronization observed in local field potentials, which is commonly used to measure arousal (*Schwartz et al., 2020*; *Vinck et al., 2015*). Thus, while this single variable is unlikely to completely characterize such a complex phenomenon as arousal, pupil provides a window into a global brain state that is activated during demanding behaviors.

The degree to which pupil size co-varies with task engagement, and whether these variables account for independent changes in neural activity, remains to be determined. To study the interaction of task engagement and pupil size on auditory neural coding, we recorded extracellular spiking activity and pupil size of ferrets during an auditory task and during passive presentation of task stimuli. Single- and multiunit activity was recorded from the primary auditory cortex (A1) and midbrain inferior colliculus (IC). We used a generalized linear model to quantify the unique contributions of task engagement and pupil size to neural activity. Using this approach, we found that independent subpopulations of neurons were modulated uniquely by task or by pupil size, suggesting that these effects are mediated by distinct inputs. Our analysis also revealed that in many neurons, pupil size could account for variability in firing rate that might otherwise be attributed to task engagement. Thus, some previously reported effects of task engagement may in fact be explained by changes in pupil-indexed arousal. In addition, the relative effects of task engagement and pupil size varied between cortex and midbrain. Pupil-related changes occurred in both areas, but task engagement effects were more prominent in A1. This work highlights the value of accounting for multiple state variables in studies of sensory coding (*Musall et al., 2019*; *Stringer et al., 2019*), both in identifying the source of task-related effects and in locating where in the sensory processing hierarchy they emerge.

## Results

### Changes in pupil size track task engagement

To study interactions between pupil-indexed arousal and task engagement, we trained four adult ferrets on a go/no-go auditory detection task (*David et al., 2012*; *Yin et al., 2010*). Animals reported the presence of a target tone following a random number of broadband noise reference stimuli (*Figure 1A,B*). They were required to withhold licking a water spout during the reference sounds and were given liquid reward for licking during the target window (0.1–1.5 s after target onset). Targets were pure tones, either presented alone (tone versus noise discrimination, ferret L) or embedded in reference stimuli (tone-in-noise detection, ferrets R, B, and T). After training, all animals performed consistently above chance (*Figure 1C*).

To track and record changes in pupil size, an infrared video camera was used to image the eye contralateral to the speaker emitting auditory stimuli (*Figure 1A*). Constant luminance was maintained throughout the experiment so that fluctuations in pupil size reflected only changes in internal state (*McGinley et al., 2015*; *Schwartz et al., 2020*). Pupil was recorded continuously as animals passively listened to the task stimuli and actively performed the task. The distribution of pupil size differed between passive and active conditions (*Figure 1—figure supplement 1A*). Mean pupil size (measured by the major axis diameter) was consistently larger during the active condition ($n = 35$ sites total, ferret L: $n = 13$, B: $n = 5$, R: $n = 16$, T: $n = 1$, $p = 0.0133$, hierarchical bootstrap, *Figure 1D*). Within active blocks, pupil size also varied with task performance. Average pupil size

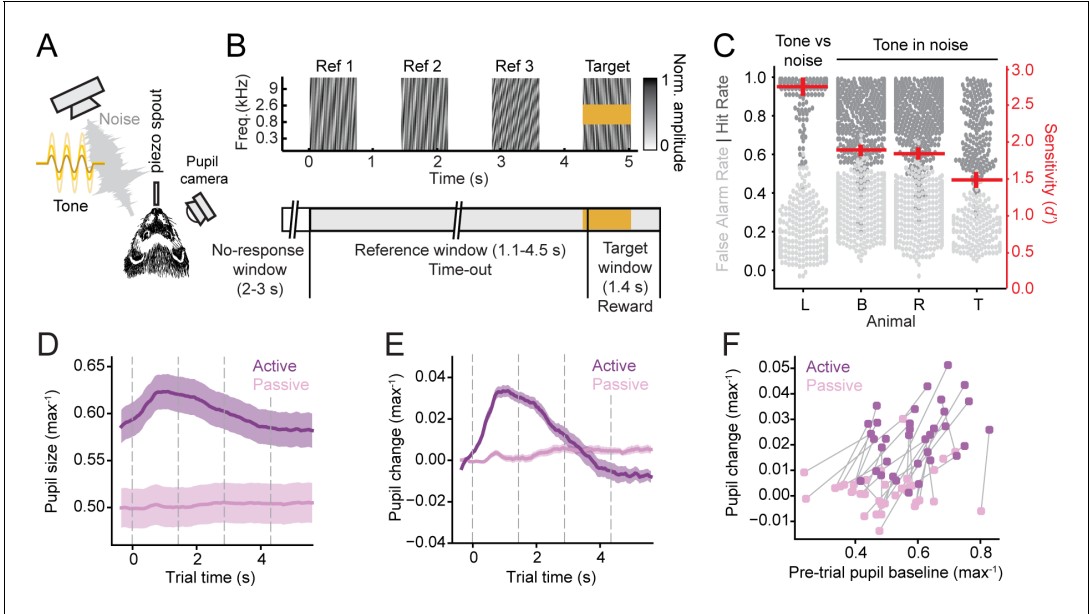

**Figure 1.** Pupil size correlates with task engagement. (**A**) Schematic of behavioral setup, including free-field speaker for sound presentation, piezo spout to register licks and deliver reward, and infrared video camera for pupillometry. (**B**) Spectrogram of example trial (top) and task structure (bottom). False alarms (FAs) before the target window were punished with a timeout and hits resulted in a liquid reward. (**C**) Swarm plot showing behavioral performance (hit rate or FA rate) for each animal. Each point corresponds to a behavioral block. Red lines indicate mean and standard error of the mean (SEM) of the sensitivity (d') for each animal. All animals performed above chance on average (d' > 1). (**D**) Mean time course of pupil size aligned to trial onset for active and passive blocks during neurophysiological recordings, averaged across all animals and recording days, normalized to maximum size per day. Shading indicates SEM. Gray dashed lines indicate sound onset. (**E**) Average time course of change in pupil size, normalized to 0.35 s pre-trial period for active and passive behavioral blocks. Shading indicates SEM. Gray dashed lines indicate sound onset. (**F**) Scatter plot compares pre-trial pupil size and mean change per trial (3 s post-onset window). Each dot represents a behavioral block. Passive and active blocks within the same session are connected by a line.

The online version of this article includes the following figure supplement(s) for figure 1:

**Figure supplement 1.** Pupil size correlates with task engagement and performance.

was similar for false alarm (FA) and hit trials, but it was smaller during miss trials, more closely resembling passive trials (*Figure 1—figure supplement 1B*). In addition, the short-term dynamics of pupil depended strongly on task condition. During behavior, trial onset evoked a rapid increase in pupil size, which was consistently larger than any evoked change during the passive condition ($p = 4.00 \times 10^{-6}$, hierarchical bootstrap, *Figure 1E,F*; *Figure 1—figure supplement 1C*). Thus, pupil size tracked slow changes in task-engagement (active versus passive blocks) as well as more rapid changes in trial-by-trial performance during the active condition.

## Distinct task engagement and pupil size effects in A1 and IC neurons

We recorded extracellular spiking activity from A1 (*n* = 129 units total, ferret B: six sites/88 units, R: one site/13 units, T: one site/28 units) and IC (*n* = 66 units total, ferret B: four sites/7 units, L: 13 sites/46 units, R: 10 sites/13 units) of ferrets, while they switched between active engagement and passive listening. In the IC, we recorded from central (ICC, *n* = 18 units) and non-central regions (NCIC, *n* = 48 units), distinguished by their functional topography (*Slee and David, 2015*). Results from the IC are presented jointly between the two sub-regions, unless otherwise specified.

To study how behavior state modulated auditory neural activity over time, we recorded activity in the same units over multiple passive and active blocks and recorded pupil size throughout each experiment as a measure of the animal's state of arousal. For analysis, the peristimulus time histogram (PSTH) response to the reference stimuli was averaged across passive and correct (hit) active trials, and data from target periods were excluded. On hit trials, animals did not lick during the reference stimuli; thus this approach reduced the possibility that lick-related motor signals confounded our results (*Brosch et al., 2005*; *Schneider et al., 2014*; *Stringer et al., 2019*).

For many units, activity appeared to be modulated by changes in task engagement, as would be expected from previous behavioral studies (*Fritz et al., 2003*; *Kuchibhotla et al., 2017*; *Niwa et al., 2012*; *Otazu et al., 2009*). During active behavior, responses to the noise stimuli either increased or decreased, and then returned to their baseline passive state during subsequent passive periods (*Figure 2A,B*). For other units, however, no consistent change in activity was observed between active and passive conditions. In these cases, firing rate could change between behavioral blocks, but the effect was not consistent across engagement conditions (*Figure 2D,E*). Thus, despite our controls for sound acoustics and motor activity, the firing rates of some units varied over time in a way that did not appear to be related to task engagement.

We wondered if changes in activity that could not be attributed to task engagement could instead be explained by fluctuations in pupil-indexed arousal (*Lin et al., 2019*; *McGinley et al., 2015*; *Schwartz et al., 2020*). A simple way to investigate if pupil size predicts changes in neural firing rate is to divide all trials (pooled across active and passive) into two groups, based on the mean pupil size, and compute a separate PSTH for each group (*Figure 2C,F*). In the example A1 units, activity was often enhanced when pupil was larger than median (large versus small pupil PSTH, *Figure 2C,F*), indicating that pupil size was positively correlated with firing rate.

Because changes in pupil size were correlated with transitions between active and passive blocks (*Figure 1E*), we could not dissociate the effects of task engagement and pupil size using just the raw PSTH. Therefore, to test our hypothesis that pupil size accounted for changes in neural activity following task engagement, we fit a generalized linear model (*Equation 1*) in which the response to

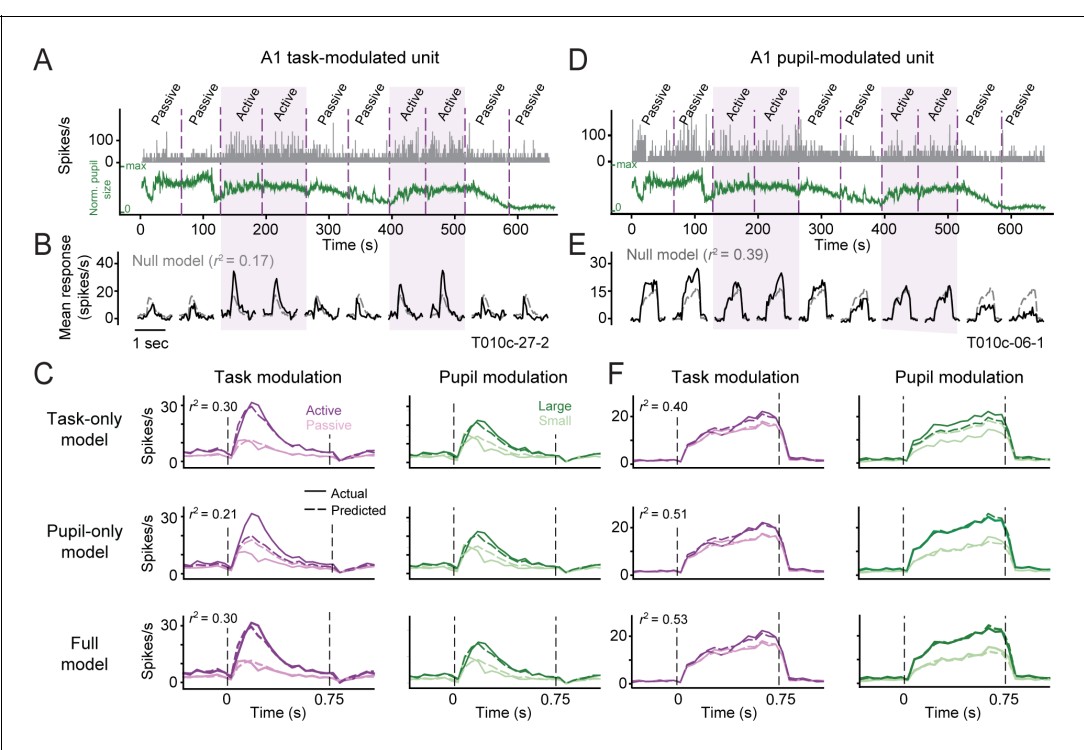

**Figure 2.** Simultaneously recorded units showing task versus pupil-related changes. (**A**) Firing rate (gray) of one unit in primary auditory cortex (A1) and time-aligned pupil size (green) during reference stimuli over one entire experiment (total recording time ~1 hr, inter-trial and target periods removed). Purple shading highlights active blocks. Dashed lines delineate boundaries between half of each behavioral block. (**B**) Peri-stimulus time histogram (PSTH) responses to reference noise averaged separately across the first and second half of each behavior block (black). Dashed gray lines are responses predicted by the null model, *i.e.*, the PSTH averaged across all blocks. Model prediction accuracy is indicated by fraction variance explained ($r^2$). (**C**) Top: PSTH response for the example unit averaged across passive and active blocks (left, dark and light purple, solid lines) and averaged across large and small pupil size trials (right, dark and light green, solid lines). Predictions of the task-only model are overlaid (same color pattern, dashed lines). Middle: Active/passive and large/small PSTH responses plotted with predictions by the pupil-only model, which only accounts for pupil-related changes. Bottom: Active/passive and large/small PSTH responses plotted with predictions by the full model, accounting for both task and pupil-related changes. (**D**) Firing rate and pupillometry for a second A1 unit, recorded simultaneously to the first example. (**E and F**) PSTH responses for the second unit, plotted as in **B and C**.

each presentation of the reference stimuli was modulated by task engagement and pupil size. Each of these variables could additively modulate the baseline (mean) firing rate or multiplicatively modulate response gain. Task engagement was modeled as a discrete regressor (active or passive) and pupil size as a continuous regressor. This approach allowed us to dissociate effects of pupil size and task engagement on firing rate.

Model performance was measured by the fraction variance explained (squared correlation coefficient, $r^2$) between the predicted and actual single-trial spike rates, using nested cross-validation. To test for significant contributions of each regressor, we introduced these variables stepwise (*Fritz et al., 2010*; *Musall et al., 2019*). The full model included both task engagement and pupil size variables. We compared performance of the full model to the performance of a state-independent model, for which both task and pupil variables were shuffled in time (null model), and of two partial models, which accounted for effects of a single state variable while shuffling the values of the other variable in time (task- or pupil partial models).

The regression analysis revealed substantial variability in effects of task engagement and pupil size on auditory responses. For one example A1 unit, the modulation of firing rate between active and passive blocks was almost completely accounted for by the task variable, with little contribution of pupil size (*Figure 2A–C*). When we incorporated the task variable, predictions significantly improved with respect to the state-independent null model (null model, $r^2 = 0.12$; task partial model, $r^2 = 0.30$; jackknifed *t*-test, $p < 0.05$; *Figure 2C*, top). Pupil size was also able to account for some changes in firing rate (pupil partial model, $r^2 = 0.21$; *Figure 2C*, middle), but the full model incorporating both task engagement and pupil size showed no additional improvement over the task partial model ($r^2 = 0.30$, $p > 0.05$; *Figure 2C*, bottom). Thus, all behavior-dependent changes for this unit were accounted for by task engagement. The apparent pupil-related effects were in fact explained by correlations between pupil size and task engagement.

Conversely, for other units, changes in firing rate were uniquely accounted for by pupil size. For a second example A1 unit (*Figure 2D–F*), the task partial model ($r^2 = 0.39$) indicated a small improvement over the null model ($r^2 = 0.40$, $p > 0.05$, jackknifed *t*-test). However, the pupil partial model showed a significant improvement over the null model, and it was able to account for changes in the PSTH between passive and active conditions ($r^2 = 0.51$, $p < 0.05$, *Figure 2F*, middle). The full model showed no additional improvement over the pupil partial model ($r^2 = 0.53$, $p > 0.05$; *Figure 2F*, bottom). Thus, we found evidence for A1 units whose activity could be modulated either by task engagement or by pupil size.

Next, we investigated the prevalence of task- and pupil-related modulation across the population of A1 and IC units. To quantify the relative contribution of each variable to model performance, we compared the variance in single-trial spike rate explained by each of the four models ($r^2$). Across the entire data set, 66/129 (51%) A1 units and 37/66 (56%) IC units showed a significant increase in $r^2$ for the full model over the null model, indicating an effect of either task engagement or pupil size ($p < 0.05$, jackknife *t*-test; *Figure 3A*). By computing the difference between the cross-validated $r^2$ for the full model and for the two partial models separately, we determined the variance that could be uniquely explained by either pupil size or task engagement in each unit. This produced four categories of state-modulated neurons (*Figure 3A*): units for which both task and pupil uniquely contributed to the modulation (black); units for which only one variable, task (purple, example unit in *Figure 2A*) or pupil (green, example unit in *Figure 2D*), contributed uniquely; and units for which changes in activity could not be uniquely attributed to either task or pupil (dark gray).

A comparison of variance explained across the entire data set shows consistent improvement by the full model over the null model in both A1 (mean $r^2$ null: 0.282, full: 0.340, $p < 10^{-5}$, hierarchical bootstrap test, *Saravanan et al., 2020*) and IC (mean $r^2$ null: 0.290, full: 0.399, $p < 10^{-5}$, *Figure 3B*). Although the proportion of units with any significant state-related modulation was comparable between A1 and IC, only about a quarter of those A1 units showed unique pupil-dependent modulation (15/66), compared to about a half for IC (20/37). The average variance uniquely explained by each state variable also differed between areas. In A1, the variance explained by task (mean $r^2$ task-unique: 0.017) was significantly greater than variance explained by pupil (mean $r^2$ pupil-unique: 0.007, $p = 0.0335$, hierarchical bootstrap, *Figure 3C*). In IC, the variance explained was not significantly different between task and pupil (mean $r^2$ task-unique: 0.016, mean $r^2$ pupil-unique: 0.019, $p = 0.769$). These effects were not significantly different between single- and multiunit data in either area (*Figure 3—figure supplement 1*). Together, these results suggest that in A1, both the

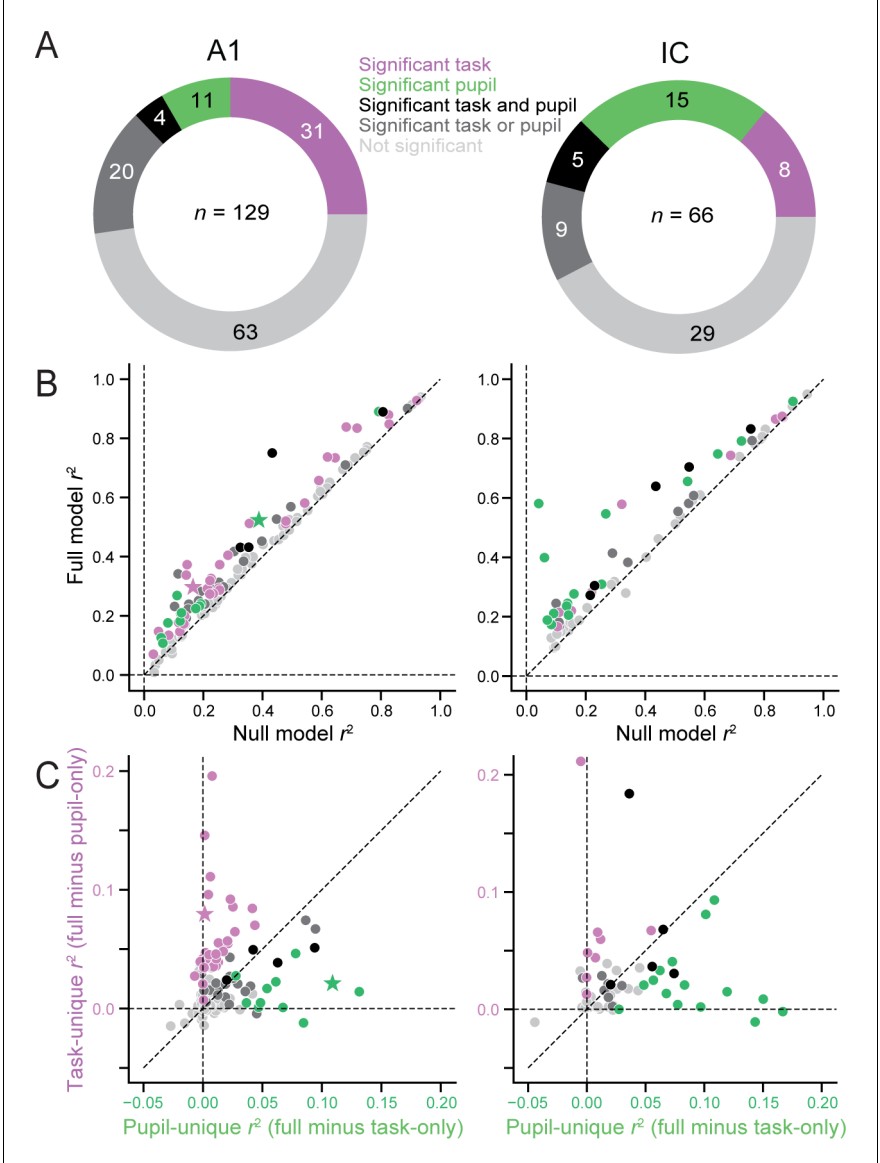

**Figure 3.** Task and pupil-related modulation of firing rates in primary auditory cortex (A1) and inferior colliculus (IC). (**A**) Doughnut plots indicate number of units significantly modulated by task engagement, pupil size, or both in A1 (left) and IC (right, $p < 0.05$, jackknifed $t$-test). Total number of recorded units reported in the center. Purple and green: significant unique modulation by task or pupil, respectively. Black: unique modulation by both task and pupil. Dark gray: ambiguous task or pupil modulation. Light gray: no significant change in accuracy between full and null models. (**B**) Scatter plot of variance explained ($r^2$) in single-trial activity by full model versus null model. Each symbol represents a unit in AC (left) or IC (right). Colors as in **A**. (**C**) Unique variance explained by pupil size (horizontal axis) plotted against unique variance explained by task engagement (vertical axis) for each unit. Stars correspond to examples in *Figure 2A* (purple) and D (green).

The online version of this article includes the following figure supplement(s) for figure 3:

**Figure supplement 1.** Unique effects of task and pupil on neural activity did not differ across single units (SU) and multiunits (MU).

magnitude and prevalence of modulation by task engagement are more pronounced than by pupil size. In IC, modulation is equal between the two state variables.

The analysis above treated task engagement as a discrete, binary variable, switching between active and passive states. However, behavioral performance, measured by $d'$, varied substantially between animals and behavior blocks (*Figure 1C*). We reasoned that performance represents a

more graded measure of task engagement and thus may explain variability in task-related changes across experiments. Indeed, we found a significant correlation between $d'$ and the unique variance explained by task in A1 ($r^2$ task-unique; $r = 0.303$, $p = 0.007$, hierarchical bootstrap; *Figure 4A*, left). The same relationship was not observed in IC ($r = 0.069$, $p = 0.318$; *Figure 4A*, right), nor was there a relationship between $d'$ and unique variance explained by pupil in either area ($r^2$ pupil-unique; A1: $r = 0.113$, $p = 0.158$; IC: $r = -0.017$, $p = 0.552$, *Figure 4B*). Thus, in A1, effects of task engagement cannot be described by a binary variable, even after dissociating effects of pupil. Instead, effects of task engagement are comprised of at least one process that varies continuously between more and less accurate behavior.

We also observed that both $d'$ and task engagement effects in A1 differed between animals (*Figure 4—figure supplement 1*). We wondered if the differences in neural modulation could be fully

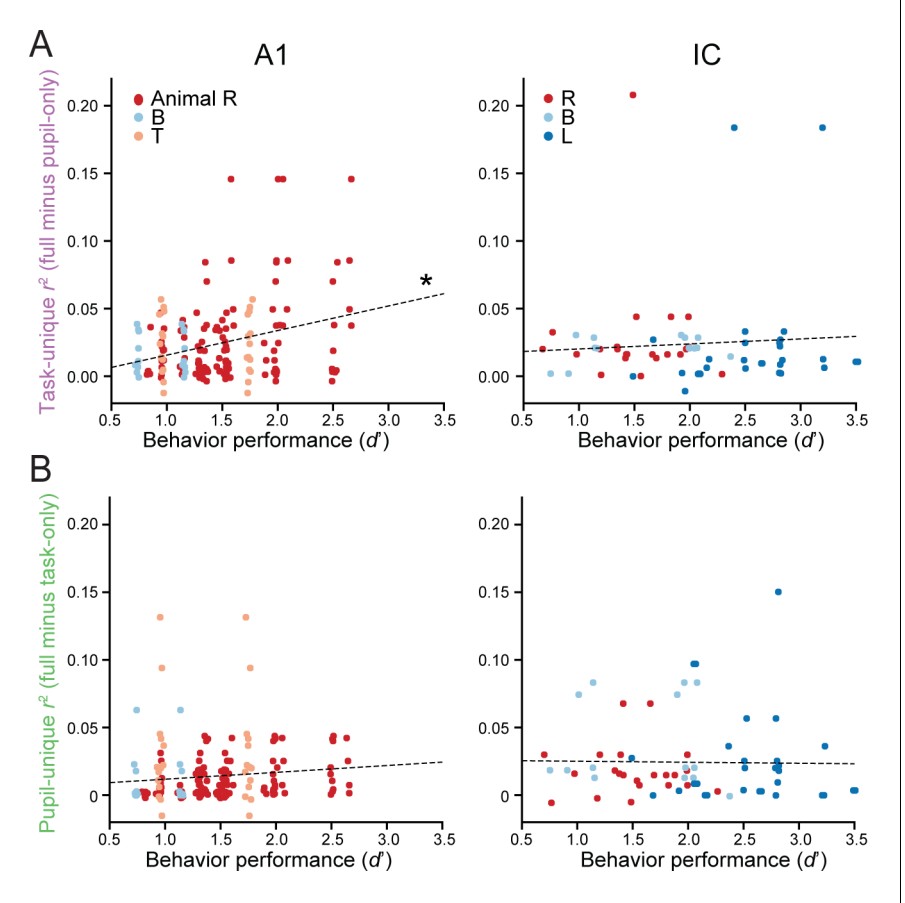

**Figure 4.** Task- and pupil-related modulation of firing rate as a function of behavioral performance in primary auditory cortex (A1) and inferior colliculus (IC). (A) Scatter plot of unique variance explained by task plotted against behavioral sensitivity ($d'$) in A1 (left) and IC (right). Each point represents a unit/target pair. When a unit was recorded across blocks with different target conditions, $d'$ and variance explained were measured separately in each block. Different colors refer to different animal subjects. Points with the same $d'$ align on the horizontal axis because they belong to experiments in which multiple units were recorded at the same time using an array (0.015 standard deviation jitter added to $d'$ to facilitate visualization). Regression analysis shows a significant correlation between unique variance explained by task and performance in A1 ($n = 132$ unit/target pairs, $r = 0.303$, $*p = 0.007$, hierarchical bootstrap) but not in IC ($n = 85$ unit/target pairs, $r = 0.069$, $p = 0.318$). (B) Scatter plot of unique variance explained by pupil plotted against performance for each unit in A1 (left) and IC (right) as in A. Regression analysis shows no significant correlation between unique variance explained by pupil and performance in either A1 ($r = 0.113$, $p = 0.158$) or IC ($r = -0.17$, $p = 0.552$).

The online version of this article includes the following figure supplement(s) for figure 4:

**Figure supplement 1.** Variability of neural behavior state effects and task performance across animals.

explained by behavioral performance or if they reflected additional between-animal differences. To answer this question, we performed a multiple regression with both $d'$ and animal identity as independent variables. This analysis revealed that in A1 $d'$ could fully explain the differences in modulation for task engagement ($d'$: $F = 16.0$, $p = 0.000093$; animal: $F = 0.66$, $p = 0.52$). Neither $d'$ or animal significantly predicted task engagement effects in IC ($d'$: $F = 1.11$, $p = 0.29$; animal: $F = 0.22$, $p = 0.80$). A similar analysis of pupil-related effects revealed that $r^2$ pupil-unique did not depend on $d'$, although it did depend on the amount of pupil variability within an experiment (*Figure 4—figure supplement 1*). Thus, differences in task engagement effects between animals could be explained by differences in the accuracy with which they performed the task.

## Pupil size accounts for apparent task engagement effects in both A1 and IC

So far, we have shown that neurons in A1 and IC show state-dependent modulation of activity that can be explained variably by task engagement and/or pupil size. The observation that pupil size is correlated with task engagement (*Figure 1*) suggests previous characterizations of task-related plasticity that did not measure pupil might have attributed changes in neural activity to task engagement when they could, in fact, be better explained by pupil size (*Downer et al., 2015*; *Fritz et al., 2003*; *Knudsen and Gentner, 2013*; *Kuchibhotla et al., 2017*; *Lee and Middlebrooks, 2011*; *Otazu et al., 2009*; *Ryan and Miller, 1977*; *Yin et al., 2014*). Therefore, we asked to what extent pupil-related modulation could explain changes in activity between active and passive blocks that would otherwise be attributed to task engagement alone.

To quantify the magnitude and sign of the modulation by task engagement, we used a modulation index ($MI_{AP}$), which measured the fraction change in mean predicted firing rate between active and passive conditions (*Equation 5*, *Otazu et al., 2009*; *Schwartz and David, 2018*). In previous work, $MI$ was measured directly from differences in the raw firing rate. However, pupil size was correlated with task engagement, and both could contribute to changes in firing rate. To tease apart their respective contributions, we used logic similar to the calculation of unique variance explained (above, *Figure 3C*). To measure task-related modulation while ignoring possible effects of pupil size, $MI$ was computed from predictions of the task only model ($MI_{AP}$ task-only). To account for effects of both task engagement and pupil size, $MI$ was computed from predictions by the full model ($MI_{AP}$ full). Modulation that could be explained by pupil size was computed using the pupil-only model ($MI_{AP}$ pupil-only). Then the unique task-related change, $MI_{AP}$ task-unique, was computed as the difference between $MI_{AP}$ full and $MI_{AP}$ pupil-only (*Equation 6*). We used a converse approach to quantify modulation uniquely attributable to pupil between large and small pupil states ($MI_{LS}$, see below).

The model predictions for the example A1 units above (*Figure 2*) illustrate the method for dissociating contributions of pupil and task engagement to $MI_{AP}$. The first example shows a strong task engagement effect that cannot be explained by pupil size ($MI_{AP}$ task-unique = 0.25, *Figure 2C*), and the second shows weak task-only modulation that can be completely explained by pupil ($MI_{AP}$ task-unique = −0.07, *Figure 2F*).

A comparison of $MI_{AP}$ task-only and $MI_{AP}$ task-unique across the entire data set showed that a task-only model often overestimated the magnitude of changes due to task engagement (*Figure 5A*). To measure this effect, we normalized the sign of $MI$ for each unit so that the means of $MI_{AP}$ task-only and $MI_{AP}$ task-unique were always positive. This normalization accounted for the bidirectionality of modulation while avoiding bias that would result from normalizing sign for just a single condition. After sign normalization, accounting for pupil size led to a decrease in $MI_{AP}$ magnitude across units in A1 and IC (A1: $p < 10^{-5}$; IC: $p = 0.0140$, hierarchical bootstrap). Effectively, accounting for pupil size led to a 33% reduction in the magnitude of $MI_{AP}$ in both areas (A1: mean magnitude $MI_{AP}$ task-only = 0.141; $MI_{AP}$ task-unique = 0.095; IC: $MI_{AP}$ task-only = 0.069; $MI_{AP}$ task-unique = 0.046). The effect of removing pupil-related activity on the magnitude of active versus passive modulation was also not different between central and non-central regions of IC (ICC versus NCIC, $p = 0.467$, hierarchical bootstrap, *Figure 5—figure supplement 1*). Taken together these results show that pupil-indexed arousal accounted for a significant portion of the change in activity between passive and active blocks in A1 and both regions of IC.

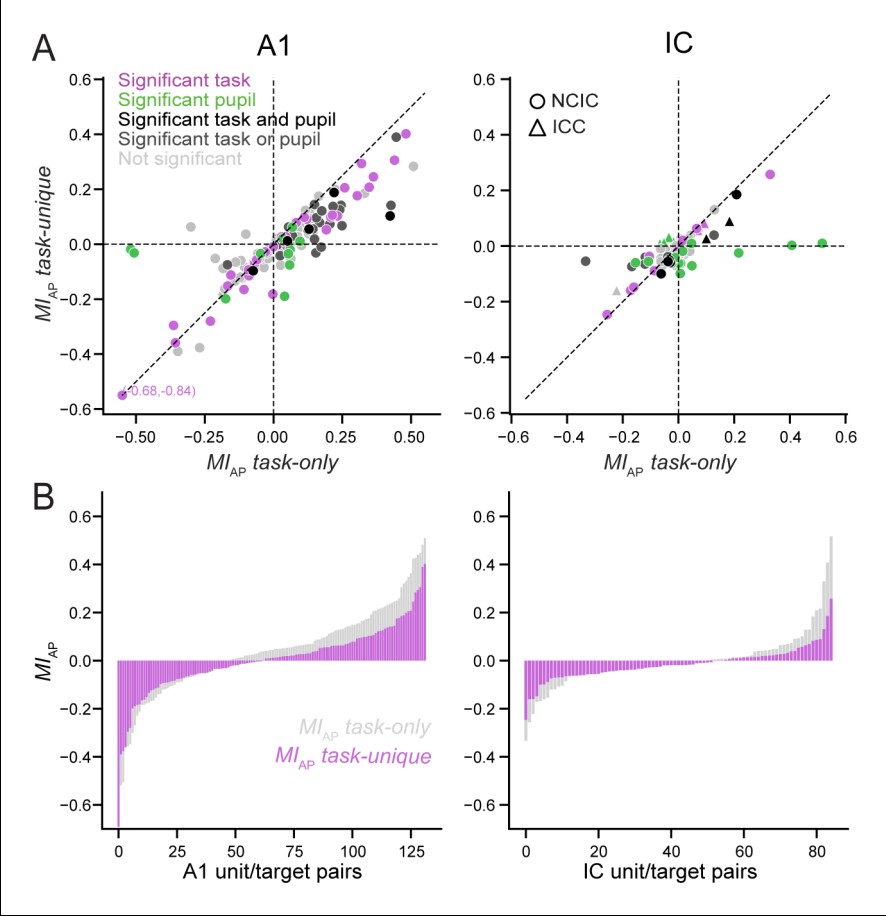

**Figure 5.** Changes in pupil size account for apparent task engagement effects. (**A**) Active versus passive modulation index, $MI_{AP}$ *task-only*, computed from responses predicted by the task only model, in which pupil size is shuffled, plotted against $MI_{AP}$ *task-unique*, in which pupil-dependent modulation is regressed out, for primary auditory cortex (A1; left, *n* = 132 unit/target pairs) and inferior colliculus (IC; right; circles for NCIC, *n* = 24 unit/target pairs, triangles for ICC units, *n* = 61). Colors indicate significant unique variance explained by one or two state variables, as in *Figure 3* (*p* < 0.05, jackknife *t*-test). (**B**) Overlaid histograms of $MI_{AP}$ *task-only* and $MI_{AP}$ *task-unique*, sorted according to their magnitude for each unit in A1 and IC. Accounting for pupil size reduced the absolute magnitude of $MI_{AP}$ by about 33% in both areas (A1: $p < 10^{-5}$; IC: *p* = 0.0140, hierarchical bootstrap). The prevalence of gray shading on the right side of the horizontal axis indicates that units with large, positive $MI_{AP}$ were most affected by this adjustment, and $MI_{AP}$ shifted to more negative values on average (A1 median $MI_{AP}$ *task-only* = 0.069, $MI_{AP}$ *task-unique* = 0.027, p = 0.0005; IC: median $MI_{AP}$ *task-only* = −0.010, $MI_{AP}$ *task-unique* = −0.037, *p* = 0.049, hierarchical bootstrap).

The online version of this article includes the following figure supplement(s) for figure 5:

**Figure supplement 1.** Task-related changes in central versus external inferior colliculus.

## Task engagement and pupil-indexed arousal modulate independent neural populations

In some units, the effects of task engagement and pupil size on firing rate were difficult to dissociate by simply looking at raw activity (*e.g.*, *Figure 2D*). However, the regression analysis could disambiguate their unique contributions and showed that often only one state variable, task or pupil, modulated the activity of individual units. Based on this observation, we wondered if effects of task engagement and pupil size operated via functionally distinct or common pathways. While identifying specific anatomical circuits was outside the scope of this work, we reasoned that if the two modulations were mediated by the same inputs to A1 and IC, a unit modulated by task engagement would also tend to be modulated by pupil size. To test this prediction, we measured the correlation between modulation that could be uniquely attributed either to task engagement or pupil. To

measure modulation by pupil, we computed changes in spike rate associated with pupil size by measuring $MI_{LS}$ (*Equation 5*), analogous to $MI_{AP}$, but dividing trials according to whether pupil size was larger or smaller than the median for the experiment, respectively. For calculation of task-unique effects, $MI_{LS}$ pupil-unique was the difference between $MI_{LS}$ full and $MI_{LS}$ task-only (*Equation 7*). Thus, we could compare the change in firing rate uniquely attributable to a change in task engagement to a change in pupil size for each unit (*Figure 6*).

We found a negative correlation between these quantities in A1 ($r = -0.281$, $p = 0.040$, hierarchical bootstrap, *Figure 6*, left) and no correlation in IC ($r = 0.104$, $p = 0.191$, *Figure 6*, right). The absence of a positive correlation between task- and pupil-related effects suggests that independent populations of neurons in A1 and IC are modulated by these variables. To test directly for dependence between task engagement and pupil size, we performed a permutation analysis, estimating the number of units that would show both task- and pupil-unique effects if the likelihood of these effects was independent ($p < 0.05$, jackknife $t$-test, counts from *Figure 3*). The frequency of joint occurrences was no greater than expected for independent distributions in both areas (A1: $p = 0.41$, IC: $p = 0.59$, permutation test). This independence is consistent with the hypothesis that separate neural mechanisms mediate effects of task engagement and pupil-indexed arousal.

Based on previous studies that used a similar positive reinforcement structure, we expected the average sign of task-related modulation to be positive in A1 (*David et al., 2012*) and negative in IC (*Slee and David, 2015*). In A1, $MI_{AP}$ task-only, computed without accounting for effects of pupil, was in fact positive on average (median 0.069, $p = 0.0240$, hierarchical bootstrap). However, $MI_{AP}$ task-unique, which accounted for pupil size, was not significantly different from zero (40/69 positive, median $MI_{AP}$ task-unique = 0.027; $p = 0.232$). In IC, average $MI_{AP}$ task-only was not different from zero ($MI_{AP}$ task-only median: $-0.010$, $p = 0.310$), but $MI_{AP}$ task-unique showed a trend toward being negative (17/49 positive, median: $-0.037$; $p = 0.062$). In contrast to the variable task-related effects, the average effect of increased pupil size was positive in A1 (55/69 positive, median $MI_{LS}$ pupil-unique = 0.0139, $p = 0.032$, hierarchical bootstrap). In IC, there was also a trend toward increased firing rate with larger pupil (31/49 positive, median $MI_{LS}$ pupil-unique = 0.022; $p = 0.090$). Thus, in

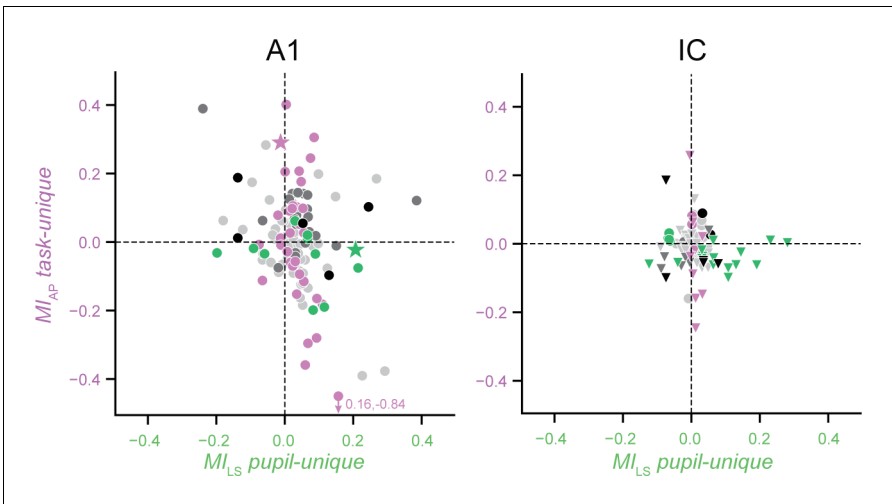

**Figure 6.** Unique task and pupil modulate independent neural populations. Scatter plot compares firing rate modulation attributed to task engagement ($MI_{AP}$ task-unique) against modulation attributed to pupil ($MI_{LS}$ pupil-unique) for each unit in primary auditory cortex (A1; left) and inferior colliculus (IC; right). The two quantities showed a weak negative correlation in A1 ($r = -0.281$, $p = 0.040$, hierarchical bootstrap) and were uncorrelated in IC ($r = -0.104$, $p = 0.191$). Colors indicate significant model performance as in *Figure 3*, and stars indicate examples from *Figure 2*.

The online version of this article includes the following figure supplement(s) for figure 6:

**Figure supplement 1.** Frequency tuning versus task-related modulation of neural activity.

**Figure supplement 2.** Task difficulty versus task-related modulation of neural activity.

both areas, accounting for higher firing rate during larger pupil led to decreased in *MI* associated with task engagement (*Figure 5*).

Some previous work has also reported that the effect of task engagement can depend on the relationship between task-relevant sound features and the receptive field of individual neurons (*David et al., 2012*; *Slee and David, 2015*). During a tone detection task, response gain in IC is selectively suppressed for neurons with best frequency (BF) near the target tone frequency (*Slee and David, 2015*). In A1, frequency tuning is suppressed at the target frequency, despite there being no systematic difference in overall response gain between units with BF near or far from the target (*David et al., 2012*). After accounting for pupil size, we observed similar trends in the data but no significant effects, likely due to the small sample size in the on-BF and off-BF groups (*Figure 6—figure supplement 1*). Task engagement effects could also depend on the difficulty or other structural elements of the task (*Carcea et al., 2017*; *Knyazeva et al., 2020*). We considered whether state-dependent effects varied with task difficulty but found no differences between pure tone, high SNR tone-in-noise (easy), and low SNR tone-in-noise (hard) conditions (*Figure 6—figure supplement 2*).

## Relationship between state modulation and auditory responsiveness

While all units were recorded from areas functionally and anatomically characterized as auditory, neurons within a single auditory field can vary substantially in their tendency to respond to sound (*Atiani et al., 2014*; *Gruters and Groh, 2012*). We noticed that some units, particularly in IC, did not show strong auditory responses but were modulated during behavior. We asked whether the magnitude of the state-dependent modulation in each recorded unit was related to how reliably it responded to repeated presentations of the same auditory stimulus. Auditory responsiveness was quantified by the variance explained by the null model, since it described how accurately the PSTH response to the reference stimuli, averaged across the entire recording, predicted responses on single trials. Larger or smaller values of null model $r^2$ were associated with units whose auditory responses were more or less reliable across the experiment, respectively. We compared this measure of auditory responsiveness to the change in $r^2$ between the full and null models, which indicates additional variance explained by the state variables. In IC, but not in A1, units that responded less reliably to sound also showed greater dependence on state (A1: $r = -0.134$, $p = 0.130$; IC: $r = -0.323$, $p = 0.0129$, hierarchical bootstrap; *Figure 7*).

These same relationships were observed when auditory responsiveness was defined as the signal-to-noise ratio (SNR) of the spectro-temporal receptive field (STRF) measured during passive listening (*Figure 7—figure supplement 1B*; *Klein et al., 2000*). The results also held true for a larger pool of neurons, which included data collected without pupillometry and analyzed with the task-only model (*Figure 7—figure supplement 1C*). High prediction accuracy by the null model limits the variance that can be explained by behavioral state (note relatively weak state effect in highest quintile in *Figure 7*). To explore how this bound might impact the analysis of auditory responsiveness, we performed a similar analysis but measured the correlation between null model performance and the fraction of remaining variance explained by pupil size and task engagement. In this case, the null model performance was positively correlated with the remaining variance explained in A1 ($r = 0.301$, $p = 0.0360$, hierarchical bootstrap) and not correlated in IC ($r = 0.124$, $p = 0.178$, *Figure 7—figure supplement 1A*). In this case the overall slope of the relationships changed, but the effects were still relatively larger for IC units with low auditory responsiveness. Thus, across multiple measures of sensory responsiveness, we observed that effects of behavior state in IC were more common in units with weak sensory responses, while in A1, state-dependent modulation was independent of sensory responsiveness.

## Changes in pupil size explain some persistent post-behavior state effects

Previous studies in both A1 (*Fritz et al., 2003*) and IC (*Slee and David, 2015*) have reported examples of task-related modulation persisting in passive blocks following active engagement (referred to as post-passive blocks). These effects have been interpreted as persistent task-related plasticity, but they are highly variable and difficult to attribute to any specific aspect of the behavior. We observed these effects in our data as well. In addition to units in A1 (*e.g.*, *Figure 2D*), some units in IC showed

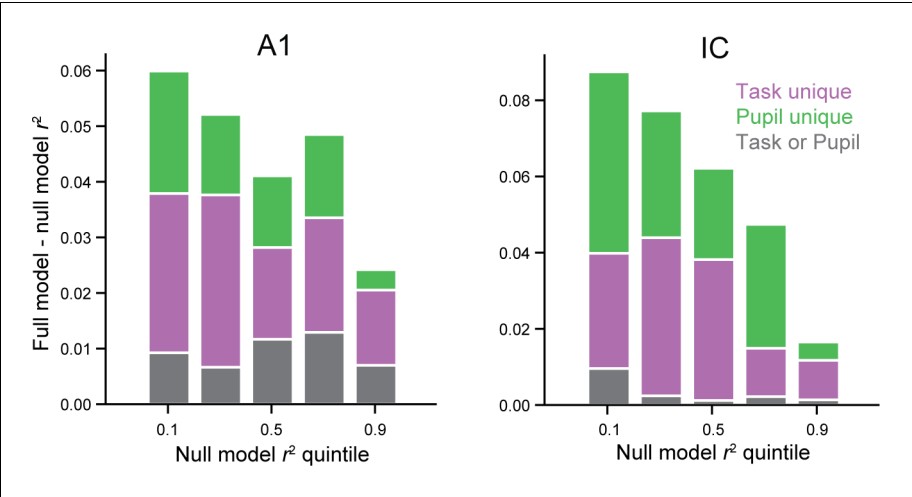

**Figure 7.** Task- and pupil-related modulation in primary auditory cortex (A1) and inferior colliculus (IC). Mean variance in firing rate explained by behavioral state variables, broken down by portions attributed to task (purple), attributed to pupil (green), and not uniquely attributable to either state variable (gray) for A1 (left) and IC (right). Data are grouped in quintiles by $r^2$ for the null model performance, a measure of auditory responsiveness. Variance explained by state was not correlated with null model performance in A1 ($r = -0.134$, $p = 0.130$, hierarchical bootstrap), but these values were negatively correlated in IC ($r = -0.323$, $p = 0.0129$). Colors as in *Figure 3*.

The online version of this article includes the following figure supplement(s) for figure 7:

**Figure supplement 1.** Auditory responsiveness predicts state-dependent modulation in inferior colliculus (IC).

persistent changes in activity during passive blocks following behavior. These changes sometimes lasted through the entire post-passive block (*Figure 8A*) and other times gradually reverted to pre-passive levels (*Figure 8B*). The time course of these changes often tracked fluctuations in pupil size.

To measure the presence of persistent task-related modulation—and the extent to which pupil size accounts for it—we compared the activity between passive blocks before (P1) and after (P2) a behavior block. We performed a regression analysis similar to that used for measuring task engagement effects, but treating passive block identity (P1 or P2) as a state variable in addition to pupil size. *MI* was computed for predictions by a model based only on block identity, ignoring pupil ($MI_{P1P2}$ block-only), and for the unique contribution of passive block after accounting for pupil effects ($MI_{P1P2}$ block-unique, *Figure 8A*). We assessed the magnitude of *MI* using the same sign normalization as for the task-unique *MI* analysis above (*Figure 5*). In both A1 and IC, the magnitude of firing rate modulation between passive blocks was significantly reduced after accounting for effects of pupil (A1: $p < 10^{-5}$, IC: $p < 10^{-5}$, hierarchical bootstrap). Accounting for pupil size led to a 20% and a 48% reduction in P1 versus P2 modulation in A1 and IC, respectively (A1: mean signed-normalized $MI_{P1P2}$ block-only = 0.140, $MI_{P1P2}$ block-unique = 0.100; IC: mean signed-normalized $MI_{P1P2}$ block-only = 0.100, $MI_{P1P2}$ block-unique = 0.052). This change was not significantly different between animals in A1 ($F = 0.669$, $p = 0.516$, one-way ANOVA) or in IC ($F = 0.446$, $p = 0.643$). While there was decrease in post-behavior effects on average, there were distinct groups of units in both A1 and IC that showed changes between P1 and P2, even after accounting for pupil size (compare units falling on the unity line versus x-axis in *Figure 8C*). Thus, while some post-behavior effects previously reported as persistent plasticity induced by behavior can in fact be explained by fluctuations in pupil size, a small number of units showed persistent post-behavior changes, even after accounting for pupil.

## Discussion

This study determined how pupil-indexed arousal contributes to task-related spiking activity in A1 and IC. Several previous studies have shown that transitioning from passive listening to active behavior leads to sizable changes in neural activity in A1 (*Fritz et al., 2003*; *Niwa et al., 2012*;

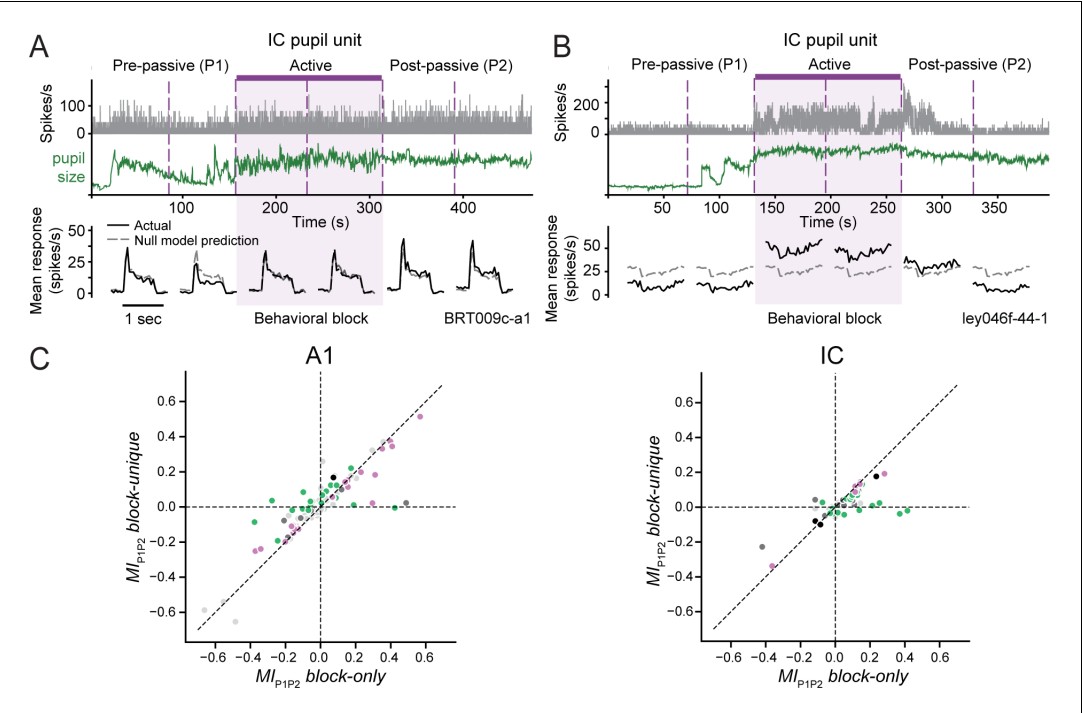

**Figure 8.** Pupil size explains some persistent task-related modulation. (**A**) Top: activity of ICC unit and concurrent pupil size recorded over passive and active blocks, plotted as in *Figure 2*. Bottom: peri-stimulus time histogram (PSTH) response to reference noise averaged over each half block (black) with null model prediction overlaid (dashed gray). The response was enhanced slightly during the post-passive (P2) block relative to pre-passive (P1). Without accounting for effects of pupil size, this suggests a sustained firing rate increase after behavior, $MI_{P1P2}$ *block-only* = 0.25. After accounting for changes in pupil, the task-related effect is reduced, $MI_{P1P2}$ *block-unique* = 0.02. (**B**) Data from example NCIC unit, plotted as in **A**. The PSTH shows a weak, suppressive response to the reference stimuli, but a large increase in mean firing rate that persists into the first half of the post-passive block ($MI_{P1P2}$ *block-only* = 0.37). Again, the persistent change in firing rate can be accounted for by the very large pupil during the first half of P2 ($MI_{P1P2}$ *block-unique* = −0.04). (**C**) Scatter plot compares $MI_{P1P2}$ *block-only* and $MI_{P1P2}$ *block-unique* for primary auditory cortex (A1; left) and inferior colliculus (IC; right). Colors as in *Figure 3*. Accounting for pupil reduces *MI* attributed to persistent effects after behavior in both areas (A1: $p < 10^{-5}$, IC: $p < 10^{-5}$, hierarchical bootstrap).

*Otazu et al., 2009*) and IC (*Ryan and Miller, 1977*; *Slee and David, 2015*). The specific changes depend on properties of the task stimuli (*Fritz et al., 2003*; *Jaramillo and Zador, 2011*; *Lee and Middlebrooks, 2011*) and on structural elements of the task, such as reward contingencies (*David et al., 2012*), task difficulty (*Atiani et al., 2009*), the degree of engagement (*Carcea et al., 2017*; *Knyazeva et al., 2020*), and the focus of selective attention (*Hocherman et al., 1976*; *Lakatos et al., 2013*; *Schwartz and David, 2018*). In the current study, we were able to use pupil size to explain changes in neural activity across both passive and active conditions. Changes in neural activity explained by pupil size are likely to account for some differences previously reported between task conditions, where different levels of arousal are required (*e.g.*, *Carcea et al., 2017*; *Knyazeva et al., 2020*; *Rodgers and DeWeese, 2014*). The effects of task engagement reported here are specific to the tone detection task and are likely to differ if task structure is changed. In contrast, pupil size explains changes in activity throughout active and passive states, and its effects are likely to be similar in different task conditions (*Schwartz et al., 2020*). By isolating aspects of behavioral state that can be explicitly attributed to changes in pupil, this study moves toward a coherent theory of how a constellation of variables interact to influence sensory coding during behavior.

Aspects of behavioral state, such as arousal, locomotion, and whisking rate, have been shown to change gradually over the course of an experiment, tracking changes in cortical spiking and performance on operant tasks (*McGinley et al., 2015*; *Niell and Stryker, 2010*; *Poulet and Petersen, 2008*). A major obstacle to understanding the relationship between behavioral state and neural activity is that state variables often covary (*Musall et al., 2019*; *Stringer et al., 2019*). To dissociate

effects of pupil size and task engagement here, we used a stepwise approach in which pupil size and task engagement provided separate regressors (*Fritz et al., 2010*; *Musall et al., 2019*). In about half of A1 and IC units, some combination of task engagement and/or pupil size explained changes in spiking activity between blocks of active engagement and passive listening. These behavioral state variables were correlated; that is, pupil size tended to increase during task engagement. However, these changes affected independent neural populations, suggesting that effects of pupil-indexed arousal and other aspects of task engagement operate via distinct feedback circuits.

## Defining meaningful state variables

The correlation between pupil size and task engagement demonstrates the importance of accounting for interactions between state variables when measuring their effects on neural activity. This observation also leads to a more fundamental question: what constitutes a meaningful measure of behavioral state? Concepts like task engagement and arousal make intuitive sense, but they become slippery when one attempts to pin them to distinct physiological processes. The current results illustrate that a commonly used binary task engagement variable is better described by a combination of pupil size and accuracy of behavioral performance. Other characterizations of task engagement have argued similarly that the degree of engagement can vary, even when relevant acoustic stimuli are held fixed (*Atiani et al., 2009*; *Carcea et al., 2017*; *Knyazeva et al., 2020*; *McGinley et al., 2015*; *Zhou et al., 2014*). Larger data sets that track continuous fluctuations in performance are likely to be able to measure fluctuations of engagement effects within behavior blocks. It remains unclear how smoothly internal state can vary. Theories of network attractors suggest that there may in fact be discrete changes in brain state associated with different behavioral contexts (*La Camera et al., 2019*). Thus, while task engagement is clearly not binary, it could still comprise multiple metastable states.

As a behavioral state variable, arousal also has multiple definitions, including the transition from wakefulness to sleep, the response to emotionally salient stimuli, and a generalized activation of the autonomic nervous system (*Satpute et al., 2019*). This relatively nonspecific behavioral concept contrasts with pupil size, which has a clear link to neurophysiology (*McGinley et al., 2015*; *Reimer et al., 2016*; *Zekveld et al., 2018*). The detailed circuits that link pupil size to cognitive state are not fully understood, but a connection to behavioral state has been demonstrated repeatedly. Thus, while pupil size may not map trivially onto the concept of arousal, it has a clear link to brain mechanisms.

A clearer understanding of the relationship between behavioral state and sensory processing may be achieved with models that incorporate additional well-defined variables like pupil size, which can be linked to well-defined neurophysiological processes. For example, motor- and reward-related activity are associated with specific neural circuits (*Bakin and Weinberger, 1990*; *Knyazeva et al., 2020*; *Schneider et al., 2014*). The current study excluded epochs that contained licking activity to avoid motor confounds, but motor signals can be included as additional model inputs that reflect aspects of signal detection and decision making (*Runyan et al., 2017*). These variables may not be directly related to an experimentally imposed task structure, but they capture processes involved in the behavior and can be quantified more concretely than abstract variables based on experimentally imposed task conditions.

## Reinterpreting previous studies of auditory behavior

Our findings indicate that changes in firing rate previously attributed to task engagement might instead reflect fluctuations in pupil-indexed arousal, particularly in IC, where pupil-related changes predominate. Changes in pupil size can also explain some modulation that persists after behavior ceases (*Fritz et al., 2003*; *Slee and David, 2015*). However, pupil size does not account for all the modulation between active and passive blocks; instead, it appears that both pupil-indexed arousal and task engagement act on independent neural populations to shape sound processing.

Previous studies that measured the effects of task engagement on auditory neuronal activity described different effects on overall excitability between A1 and IC. In A1 the effects of task engagement were shown to be enhancing (*David et al., 2012*), while in IC these changes were predominantly suppressive (*Slee and David, 2015*). Separate studies have shown that pupil size is positively correlated with excitability in both A1 and IC (*Joshi et al., 2016*; *Schwartz et al., 2020*). Here

we also found that increased pupil size enhanced excitability in both areas. However, after accounting for pupil-related changes, task-engagement effects were no longer enhancing in A1, and they were more suppressive in IC. Pupil-related changes did not depend on tuning for the target tone. These nonspecific effects of pupil size contrasted with the effects of task engagement, which can be selective for task-relevant features (*Fritz et al., 2003*; *Kuchibhotla et al., 2017*; *Slee and David, 2015*).

In the current study, significant state-dependent A1 units were equally likely to be modulated by task engagement, by pupil size, or by both. In IC, however, we found that a larger portion of state-dependent changes could be explained by pupil size. Furthermore, in IC but not in A1, units with less reliable auditory responsiveness were those with a stronger state-modulation component. This result is perhaps not surprising given that the majority of IC units in our sample were recorded from non-central regions of the IC (NCIC). These are dorsal and lateral regions of IC that receive feedback from cortex (*Winer, 2006*), input from neuromodulatory nuclei such as the pedunculopontine and latero-dorsal tegmental nuclei (*Motts and Schofield, 2009*), as well as multisensory information from somatosensory, visual, and oculomotor centers (*Gruters and Groh, 2012*). These results suggest that the auditory midbrain is not a passive sound processor. Instead, the network of midbrain nuclei in the IC plays an active role enhancing behaviorally relevant signals. Experiments that sample more units across the different subregions would be needed to confirm the interplay of behavioral and sensory signals across IC.

Multiple regression has been used previously to dissociate effects of pupil size and movement on neural activity (*Musall et al., 2019*; *Stringer et al., 2019*). This study used this approach to dissociate the effects of state-related variables and differentiate where effects emerge. Several other variables have been shown to impact auditory responses during behavior, including selective attention, temporal expectation, reward value, and motor responses (*David et al., 2012*; *Huang et al., 2019*; *Jaramillo and Zador, 2011*; *Lakatos et al., 2013*; *Metzger et al., 2006*). The same method can be adopted to explore the contribution of other state variables measured in normal conditions as well as in perturbation experiments to identify the circuits through which such variables operate.

### The role of neuromodulation in effects of task engagement and pupil-indexed arousal in A1 and IC

The correlation between task engagement and pupil size may help explain a number of results related to neuromodulation in the auditory system. It is well established that neuromodulators can induce short-term changes in activity and sensory tuning in cortex (*Bakin and Weinberger, 1996*; *Goard and Dan, 2009*) and midbrain (*Gittelman et al., 2013*; *Habbicht and Vater, 1996*; *Hurley and Pollak, 2005*). Cholinergic fibers projecting to A1 from the nucleus basalis play a key role in rapid switching between passive listening and active engagement by modulating the activity of different populations of cortical inhibitory interneurons (*Kuchibhotla et al., 2017*). In addition, the activity of cholinergic and noradrenergic terminals in A1 was found to be elevated during pupil dilation and reduced during pupil constriction (*Reimer et al., 2016*). Thus, these neuromodulatory systems have been implicated in effects of both task engagement and pupil-indexed arousal. The correlation between these state variables could be explained by shared cholinergic and noradrenergic signaling.

## Materials and methods

All procedures were approved by the Oregon Health and Science University Institutional Animal Care and Use Committee (protocol IP1561) and conform to National Institutes of Health standards.

### Surgical procedure

Animal care and procedures were similar to those described previously for neurophysiological recordings from awake behaving ferrets (*Slee and David, 2015*). Four young adult male ferrets were obtained from an animal supplier (Marshall Farms, North Rose, NY). After a 2-week quarantine, a sterile surgery was performed under isoflurane anesthesia to mount two head-posts for head fixation and to permit access to auditory brain regions for recordings. A UV light-cured composite (Charisma, Heraeus Kulzer International) was used to mount two custom-made stainless-steel head-posts spaced approximately 1 cm apart along the sagittal crest of the skull. The stability of the implant

was achieved using 8–10 stainless steel bone screws mounted in the skull (Depuy Synthes, Raynham, MA), which were covered along with the headpost in additional Charisma and acrylic denture material (Co-oral-ite Dental, Diamond Springs. CA). Two 1.2 × 1.2 cm wells over A1 in the implant were covered with only a thin layer of Charisma to allow access to auditory regions.

Following surgery, animals were treated with prophylactic antibiotics (Baytril 10 mg/kg) and analgesics (Buprenorphin 0.02 mg/kg) under the supervision of University veterinary staff. For the first 2 weeks the wound was cleaned with antiseptics (Betadine 1:4 in saline and Chlorhexidine 0.2%) and bandaged daily with application of topic antibiotic ointment (Bacitracin). After the wound margin was healed, cleaning and bandaging occurred every 2–3 days throughout the life of the animals. This method revealed to be very effective in minimizing infections of the wound margin. After recovery (~2 weeks), animals were habituated to a head-fixed posture inside the sound-booth chamber for about 2 weeks prior to the beginning of the training.

## Behavioral paradigm and training

The animals were trained by instrumental conditioning to perform a positively reinforced, tone versus noise discrimination task (ferret L) (*Slee and David, 2015*) or tone-in-noise detection task (ferrets R, B, and T). During training, animals were provided access to water ad libitum on weekends, but were placed on water restriction during the weekdays (Monday through Friday), allowing them to maintain >90% of their baseline body weight long term. On weekdays, they were given the opportunity to receive liquid reward during behavioral training. Each behavioral trial consisted of a sequence of two to five broadband noise reference sounds (TORCs; 30 samples, five octaves, 0.75 s duration, 0.7 s inter-stimulus interval) (*Klein et al., 2000*), followed by a target tone, either alone (tone versus noise task) or overlapping with another TORC (tone-in-noise task). Animals reported the occurrence of the target by licking a water spout (*Figure 1A*). Licks were detected by a piezoelectric sensor glued to the spout. Licks occurring within the target window, 0.1–1.5 s after target onset, were rewarded with one to three drops of a 2:1 solution of water and a high-protein, high-calorie dietary supplement (Ensure). FAs, licks occurring before the target window, and misses, a failure to lick during the target window, resulted in a 5–8 s timeout. Regardless of trial outcome, the next trial began only after animals did not lick for a random period (2.5 ± 0.5 s), reducing the prevalence of accidental, short-latency FAs. The number of TORCs per trial was distributed randomly with a flat hazard function to prevent behavioral timing strategies (*Heffner and Heffner, 1995*). Behavioral performance was measured using $d'$, the z-scored difference between hit rate (HR) and FA rate (*Green and Swets, 1966*).

A behavioral block consisted of 60–100 trials with the same target tone frequency (100–20,000 Hz) and same distribution of target SNR. Animals completed one to three behavioral blocks per day, between which target frequency and/or SNR could change. During training, target frequency was chosen at random to span the audiogram of a ferret (*Kelly et al., 1986*). During electrophysiological recordings, target tone frequency was selected to match the BF of the recording site.

During the training on the tone-in-noise variant of the task, the TORC sample masking the target varied randomly in each trial to prevent animals from using TORCs' spectro-temporal features to identify targets. At the beginning of training, the tone was presented at +40 dB SNR (ratio of peak-to-peak amplitude) relative to the TORCs. This difference was gradually reduced over the course of 2–3 weeks until the animal consistently performed above chance (three behavioral blocks with performance yielding $d'$>1, see below) at 0 dB SNR.

For the tone-in-noise task, the SNR of the tone with respect to the overall level of the TORCs (fixed at 55 or 60 dB SPL depending on the animal) varied between +5 and −20 dB SNR, in 5 dB steps. Each session included five target/noise SNRs. To manipulate task difficulty within each session, the probability of each of the five target/noise SNRs varied, yielding two difficulty conditions: a high SNR ('easy') condition in which 60% of the trials the target/noise SNR was either the highest or the second to the highest SNR within the target/noise SNR distribution; and a low SNR ('hard') condition in which the two lowest target/noise SNRs occurred in 60% of the trials. For example, for ferret B, the distribution after training was completed and kept between 0 and −20 dB SNR, so that in the easy condition 0 and −5 dB SNR targets would appear 60% of the time, −10 dB SNR 20% of the time, and the remaining −15 and −20 dB SNR would be presented another 20% of the trials. During electrophysiological tone-in-noise experiments, the tone was embedded in a single TORC sample,

which also occurred in the reference period. We confirmed that animals were not biased to respond to this TORC exemplar in the reference phase.

## Sound presentation

Behavioral training and subsequent neurophysiological recording took place in a sound-attenuating chamber (Gretch-Ken) with a custom double-wall insert. Stimulus presentation and behavior were controlled by custom MATLAB software (code available at https://bitbucket.org/lbhb/baphy). Digital acoustic signals were transformed to analog (National Instruments), amplified (Crown), and delivered through two free-field speakers (Manger, 50–35 000 Hz flat gain) positioned ±30˚ azimuth and 80 cm distant from the animal. Stimuli were presented either from the left or the right speaker, contralaterally to the recording site. Sound level was equalized and calibrated against a standard reference (Brüel and Kjær).

## Pupil recording

During experiments, infrared video of one eye was collected for offline measurement of pupil size as an index of arousal (pupil-indexed arousal). Ambient light was maintained at a constant level to prevent light-evoked changes in pupil and to maximize the dynamic range of pupil-related changes (*Schwartz et al., 2020*). Recordings were collected using a CCD camera (Adafruit TTL Serial Camera 397) fitted with a lens (M12 Lenses PT-2514BMP 25.0 mm) whose focal length allowed placement of the camera 10 cm from the eye. To improve contrast, the imaged eye was illuminated by a bank of infrared LEDs. Ambient luminance was provided using a ring light (AmScope LED-144S). At the start of each recording day, the intensity of the ring light was set to a level (~1500 lux measured at the recorded eye) chosen to give a maximum dynamic range of pupil sizes. Light intensity remained fixed across the recording session. Pupil size was measured from the video signal offline using custom MATLAB software, which is detailed in *Schwartz et al., 2020*.

## Neurophysiology

Data were recorded during behavior and pupillometry from 129 units in A1 and 66 units in IC. For one supplementary analysis (*Figure 7—figure supplement 1*), additional data were collected during behavior but without pupillometry (IC: 135 additional units, A1: 125 additional units). After animals demonstrated consistent above-chance performance (target versus reference $d' > 1$, see below), a small craniotomy was opened to access either A1 or IC. Extracellular neuronal activity was recorded in non-anesthetized ferrets either using a tetrode (Thomas Recording Inc) or a linear 64-channel silicon probe (*Shobe et al., 2015*). The impedance of the tetrode was measured to be 1–2 MOhm, and the 64-channel probe was electroplated to reach a 0.7-MOhm impedance in each channel. The tetrode and probe were inserted using a motorized electrode positioning system (Alpha-Omega).

Amplified (tetrodes, AM Systems; arrays, Intan) and digitized (National Instruments) electrophysiological signals were stored using the open-source data acquisition software MANTA (*Englitz et al., 2013*) or Open Ephys (*Black et al., 2017*). Recording sites in A1 and IC were initially targeted stereotactically (*Moore et al., 1983*; *Shamma et al., 1993*). Sites in A1 were confirmed based on tonotopy—which changes from high to low BF as one moves from dorso-medial to ventro-lateral regions—and relatively reliable and simple frequency tuning (*Shamma et al., 1993*). Recording sites in the IC were classified based on tonotopy and response properties (*Aitkin et al., 1975*; *Moore et al., 1983*). Neurons in the central nucleus of the IC (here referred to as ICC) receive input from the auditory brainstem, and have a characteristic short response latency, dorsal-ventral tonotopy, and narrow bandwidth tuning. Conversely, regions around the central nucleus do not receive direct ascending input and present longer response latencies, considerably less sharp tuning, lack consistent tonotopic organization; these areas were grouped together as NCIC (*Slee and David, 2015*).

For tetrode recordings, upon unit isolation, a series of brief (100 ms duration, 200–400 ms interstimulus interval, 50 dB SPL) tones and/or narrowband noise bursts were used to determine the range of frequencies that evoked the strongest response, the BF of the unit(s). If a unit(s) in the site did not respond to the sound stimuli (an evoked increase or decrease in activity compared to spontaneous activity either during or after sound presentation), the electrode was moved to a new

recording depth with small, 5 μm incremental steps. For the 64-channel recordings, we lowered the probe so that its entire extent (1 mm) spanned the target region (A1 or IC).

## Spike sorting

Putative spikes were sorted offline by band-pass filtering the raw trace (300–6000 Hz) and extracting events from the continuous signal that deviated $\geq 4$ standard deviations from zero. To separate single units and stable multiunits from the electrode signal, we used the Catamaran clustering program (kindly provided by D. Schwarz and L. Carney) (*Schwarz et al., 2012*) for tetrode recordings and the software KiloSort (*Pachitariu et al., 2016*) for array recordings. In both cases, units were defined based on visual inspection of traces and by having less than 1% of inter-spike intervals shorter than 0.75 ms (single units) and less than 2.5% inter-spike intervals shorter than 0.75 ms (multiunits). Stability of single-unit isolation was verified by examining waveforms and interval histograms. If isolation was lost during an experiment, only activity during the stable period was analyzed.

## Analysis

Data preprocessing and analysis were performed using custom MATLAB and Python scripts. All analyses were performed on biological replicates, either different behavior sessions, different neurons (single- or multiunit, see above), or different neuron-behavior condition pairs. Neural and pupil activity were binned at 20 samples/s before analysis. A Python library used for the modeling portion of this analysis is available at https://github.com/LBHB/NEMS/. Sample size (number of animals, number of units per condition) was determined based on norms established in previous studies of neural activity in awake, behaving animals (*Fritz et al., 2003*; *Niwa et al., 2012*; *Otazu et al., 2009*).

## Task performance

Behavioral performance was measured using signal detection theory (*Green and Swets, 1966*; *Stanislaw and Todorov, 1999*). Hits or *FAs* occurred when the animal licked the piezo waterspout upon presentation of the target or the reference stimuli, respectively. Misses or correct rejections (*CRs*) occurred when the animal did not lick following presentation of the target or reference. *HR* was calculated as the proportion of licks that occurred upon presentation of the target during the response window (0.1–1.5 s after target onset), *HR = hits/(hits+misses)*. FA rate (*FAR*) was calculated as the proportion of reference stimuli that elicited a lick, during the same window following reference onset, *FAR = FAs/(FAs + CRs)*. Sensitivity (*d′*), a measure of the animals' ability to discriminate between target tone and reference noise, was measured by the difference between the z-scored HR and the FA rate, $d' = HR_z - FAR_z$. No discrimination, that is, equal likelihood of responding to target and reference is indicated by $d' = 0$. Animals were considered trained to the task and ready to undergo electrophysiological recordings when they performed consistently above chance ($d' > 1$) for three consecutive sessions.

## Auditory tuning properties

Tuning properties of A1 and IC units were characterized using the STRF estimated by reverse correlation between time-varying neuronal spike rate and the rippled noise used as reference sounds during behavior (*Klein et al., 2000*). The STRF provides a quick and efficient measure of stimulus frequencies that enhance or suppress neural firing. In the current study, the STRF was used to measure BF as the center of mass of a spectral tuning curve computed from the first principle component of the STRF matrix (*Simon et al., 2007*). Auditory responsiveness was computed as the SNR of single trial responses to the reference noise stimuli during passive listening (*Klein et al., 2000*).

## State-dependent models

We fit a generalized linear model in which time-varying state variables pupil size, $p(t)$, and task engagement, $b(t)$, were used to re-weight each unit's mean evoked response to each noise stimulus, $r_0(t)$, and spontaneous rate, $s_0$, to generate a prediction of the single-trial spike rate, $r_{full}(t)$, at each point in time. The model included multiplicative gain parameters ($g$) and DC offset parameters ($d$) to capture both types of modulation. We refer to this model as the *full model*:

$$r_{full}(t) = s_0 F_d \big[ d_0 + d_p p(t) + d_b b(t) \big] + r_0(t) F_g \big[ g_0 + g_p p(t) + g_b b(t) \big] \tag{1}$$

To account for nonlinear threshold and saturation of state effects, the summed state signal was passed through a sigmoid nonlinearity, $F_d$ or $F_g$, before scaling response baseline or gain, respectively (difference of exponentials) (*Thorson et al., 2015*). With additional constant terms $g_0$ and $d_0$, this model required a total of six free parameters. For comparison, we calculated a state-independent model here referred to as the *null model*, in which the state variable regressors were shuffled in time. Because shuffling removes any possible correlation between state and neural activity, gain and offset parameters are reduced to $d_0=g_0=1$ and $d_p=d_b=g_p=g_b=0$, effectively reducing the model to

$$r_{null}(t) = s_0 + r_0(t) \qquad (2)$$

In practice, fitting to the shuffled data produces parameter values slightly different from zero, and controls for noise in the regression procedure.

We also considered two *partial models*, one to predict responses based on pupil size only, $r_p(t)$, and the other to predict responses based on behavior only, $r_b(t)$, in which the other regressor was shuffled in time. Thus, the pupil-only model accounted only for effects of pupil size,

$$r_p(t) = s_0 F_d \big[d_0 + d_p p(t)\big] + r_0(t) F_g \big[g_0 + g_p p(t)\big] \qquad (3)$$

and the task-only model accounted only for task engagement,

$$r_b(t) = s_0 F_d [d_0 + d_b b(t)] + r_0(t) F_g [g_0 + g_b b(t)] \qquad (4)$$

These models tested the effects of a single state variable while ignoring the other.

By comparing performance of the full model to each partial model, we determined the unique contribution of each state variable to the neuron's activity. We used a 20-fold, nested cross-validation procedure to evaluate model performance, which permitted using the full data set for both model fitting and validation without introducing bias from over-fitting. The model was fit to 95% of the data and used to predict the remaining 5%. Fit and test data were taken from interleaved trials. This procedure was repeated 20 times with nonoverlapping test sets, so that the final result was a prediction of the entire response. Model performance was then quantified by the fraction of variance explained, that is, the squared correlation coefficient, $r^2$, between the predicted and actual time-varying response. Variance uniquely explained by single state variables was calculated as the difference between $r^2$ for the full model and for the partial model in which the relevant variable was shuffled in time.

When comparing pupil and neural data, a 750 ms offset was applied to pupil trace to account for the lagged relationship between changes in pupil size and neural activity in A1 (*McGinley et al., 2015*).

## Modulation Index

To quantify the modulatory effects of task and pupil size on the firing rate of A1 and IC units, we computed a modulation index, *MI* (*Otazu et al., 2009*; *Schwartz and David, 2018*). The sound-evoked response was computed by averaging firing rate over the 750 ms duration of a stimulus presentation. *MI* was defined as the difference between the mean response to the same stimulus in two conditions, $\alpha$ and $\beta$, normalized by the sum,

$$MI_{\alpha\beta} = \frac{r_\alpha - r_\beta}{r_\alpha + r_\beta} \qquad (5)$$

*MI* could be calculated between different behavioral blocks or between state conditions. In the case of task engagement, *MI* was calculated between active and passive conditions, $MI_{AP}$. For pupil-dependent changes, data from an experiment were divided at the median pupil diameter, and $MI_{LS}$ was computed between large pupil (above median, high arousal) and small pupil (below median, low arousal). To quantify the differences between the first and the second passives, $MI_{P1P2}$ was computed between the first and second passive blocks.

To quantify the changes in firing rates due to unique contributions of task condition or pupil size, we used *MI* to test how well the regression model could predict state-related changes in neural activity. The modulation between conditions $\alpha$ and $\beta$ predicted by the full model is denoted as $MI_{\alpha\beta}$ *full*, where $\alpha$ and $\beta$ are either active/passive ($MI_{AP}$ *full*) or large/small pupil ($MI_{LS}$ *full*). Similarly,

modulations between conditions $\alpha$ and $\beta$ predicted by the pupil partial model or the behavior partial model are denoted $MI_{\alpha\beta}$ *pupil-only* and $MI_{\alpha\beta}$ *task-only*, respectively. The *MI* uniquely predicted by including task engagement as a regressor is

$$MI_{AP}\,task\,unique = MI_{AP}\,full - MI_{AP}\,pupil\,only \tag{6}$$

that is, *MI* predicted by the full model minus *MI* predicted by a model in which behavior condition, but not pupil, is shuffled. The net result is the *MI* predicted by task engagement above and beyond modulation predicted by changes in pupil size alone. Similarly, *MI* uniquely predicted by including pupil size as a regressor is

$$MI_{LS}\,pupil\,unique = MI_{LS}\,full - MI_{LS}\,task\,only \tag{7}$$

Significant effects of regressing out pupil size were quantified by comparing the signed-normalized differences between $MI_{AP}$ *task-unique* and $MI_{AP}$ *task-only* and differences of the same quantities across areas were quantified using a hierarchical bootstrap test (*Saravanan et al., 2020*). Sign normalization was achieved by multiplying the difference between $MI_{AP}$ *task-unique* and $MI_{AP}$ *task-only* in each unit by the sign of their mean.

Significantly modulated units were determined by comparing Pearson's *R* coefficients associated with the full model and with the difference between the full model and the task and pupil partial models using jackknifed *t*-test with $\alpha$ = 0.05 (*Efron and Tibshirani, 1986*).

## Statistical tests

Several analyses assessed changes in the median of repeatedly sampled data, *e.g.*, average pupil size measured across multiple behavior blocks (*Figure 1*) or average state-dependent modulation measured across multiple units (*Figures 3* and *5–8*). In this case, significant differences were assessed using a hierarchical bootstrap test (*Saravanan et al., 2020*). The hierarchical bootstrap is nonparametric, like the more traditional Wilcoxon signed-rank test, but it accounts for potential bias resulting from the relatively small number of array recordings in the data set. A statistical test that treats each unit as an independent measure could potentially be biased if a single array recording produced spuriously large effects, and the hierarchical bootstrap controls for this possibility. The bootstrap analysis was run for 10,000 iterations, so that the minimum measurable *p*-value was $10^{-5}$. Thus, results that returned *p* = 0 after this many iterations are reported as $p < 10^{-5}$.

State-dependent changes in individual neurons were assessed using a combination of nested cross-validation and a jackknife *t*-test (see above, *Efron and Tibshirani, 1986*). To determine if any population-level effects depended on task performance or between animal differences, we used multivariate ANOVA (*Figure 4*).

## Acknowledgements

This study was supported by NIH grants F31 DC014888 (DS), F31 DC016204 (ZPS), R01 EB028155 (SVD) and R01 DC0495 (SVD), by NSF grant GVPRS0015A2 (CRH) and by the Tartar Trust at Oregon Health & Science University (DS).

## Additional information

### Funding

| Funder | Grant reference number | Author |
| --- | --- | --- |
| National Institutes of Health | F31 DC014888 | Daniela Saderi |
| National Institutes of Health | R01 DC04950 | Stephen V David |
| National Institutes of Health | R01 EB028155 | Stephen V David |
| National Institutes of Health | F31 DC016204 | Zachary P Schwartz |
| National Science Foundation | GVPRS0015A2 | Charles R Heller |
| Oregon Health and Science | Tartar Trust Fellowship | Daniela Saderi |

University

The funders had no role in study design, data collection and interpretation, or the decision to submit the work for publication.

## Author contributions
Daniela Saderi, Conceptualization, Data curation, Formal analysis, Funding acquisition, Investigation, Visualization, Methodology, Writing - original draft, Writing - review and editing; Zachary P Schwartz, Jacob R Pennington, Software, Writing - review and editing; Charles R Heller, Software, Formal analysis, Visualization, Writing - review and editing; Stephen V David, Conceptualization, Software, Supervision, Funding acquisition, Methodology, Writing - original draft, Writing - review and editing

## Author ORCIDs
Daniela Saderi https://orcid.org/0000-0002-6109-0367
Stephen V David https://orcid.org/0000-0003-4135-3104

## Ethics
Animal experimentation: All procedures were approved by the Oregon Health and Science University Institutional Animal Care and Use Committee (protocol IP1561) and conform to National Institutes of Health standards.

## Decision letter and Author response
Decision letter https://doi.org/10.7554/eLife.60153.sa1
Author response https://doi.org/10.7554/eLife.60153.sa2

# Additional files
## Supplementary files
• Transparent reporting form

## Data availability
Neurophysiology data is available via Zenodo. Software used for analysis is available via GitHub.

The following dataset was generated:

| Author(s) | Year | Dataset title | Dataset URL | Database and Identifier |
|---|---|---|---|---|
| Saderi D, Schwartz ZP, Heller CR, Pennington JR, David SV | 2021 | Dissociation of task engagement and arousal effects in auditory cortex and midbrain – dataset. | https://zenodo.org/record/4437077 | Zenodo, 10.5281/zenodo.4437077 |

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
