## [Decision Letter]

**Acceptance summary:**

This study distinguishes effects of generalized arousal and specific task engagement on the activity of neurons in the inferior colliculus and auditory cortex of ferrets as they engaged in a tone detection task while monitoring arousal via pupillometry. The authors found that arousal effects were more prominent in IC, while arousal and engagement effects were equally likely in A1. Task engagement was correlated with increased arousal. The authors propose that there is a hierarchy such that generalized arousal enhances activity in the midbrain, and task engagement effects are more prominent in cortex, showing that the effects of brain state and behavioral context are area specific.

**Decision letter after peer review:**

Thank you for submitting your article "Dissociation of task engagement and arousal effects in auditory cortex and midbrain" for consideration by *eLife*. Your article has been reviewed by three peer reviewers, including Catherine Emily Carr as the Reviewing Editor and Reviewer #1, and the evaluation has been overseen by Andrew King as the Senior Editor.

The reviewers have discussed the reviews with one another and the Reviewing Editor has drafted this decision to help you prepare a revised submission.

Summary

Saderi and her colleagues have attempted to determine whether and how two behavior-related variables – arousal and task engagement – differently influence activity in two stages of the auditory neuraxis, IC and A1. They define arousal as pupil diameter and task-engagement as a binary variable determined by the experimental block design. They find that although these two parameters often co-vary, they sometimes do not. They find that IC was more influenced by arousal and A1 was modulated by both arousal and engagement. One of their main findings is that previous reports of task-engagement effects may in fact be attributed to arousal state.

Essential revisions

1) The major concern is the use of a continuous readout of arousal (i.e. pupil diameter) with a binary readout of task-engagement (i.e. the block the animal is in at any moment). The reviewers would like to know if task engagement be explained more rigorously as a continuous rather than binary variable. One notes that when training and testing animals on appetitive behaviors, task engagement can wax and wane within a single block, across an experimental recording session, or across days of behavioral testing. Such changes in engagement can be inferred, for example, as strings of (seemingly) easy trials in which the animal does not answer correctly. The authors should attempt to quantify through behavioral analysis (running lapse rate, lick latency, etc) whether and how task engagement may be changing within and across task blocks. Alternatively, the authors could clearly explain that their binary encoding of engagement has limitations and may not actually describe the animal's engagement at any given moment.

2) Can a continuous readout of task engagement better explain neural activity? For many neurons, task-engagement does not provide unique predictive information, yet for others it does (e.g. Figure 3C). If task engagement can be modeled as a continuous rather than binary variable, is it still true that "some apparent effects of task engagement should in fact be attributed to fluctuations in arousal" (Abstract)? There is a concern that the current analysis maybe a floor on task-related modulations since it assumes constant engagement throughout a task block.

3) Can neural heterogeneity be attributed to animal-to-animal behavioral variability? Even if task engagement does not vary within a task block for any one animal, it may indeed vary across animals. In theory, the actual task engagement of some animals might more closely mirror the block design that the experimenters are imposing, and some animals may simply have a higher level of engagement than others. This could mean that some results that are currently attributed to population-level heterogeneity (e.g. some A1 neurons do this, while others do that) might actually be attributed to animal-to-animal heterogeneity as opposed to distinct neural populations. For example, the authors state that for a subset of neurons, persistent task-like activity after a block change can be accounted for by pupil size, whereas for other neurons this effect cannot (Figure 7). The authors should confirm that key findings are consistent across animals and not related to degrees of task engagement (see point #1). If the findings are not consistent across animals but can be explained by each animals' unique behavior, this would also be really cool.

4) With respect to recordings, the authors should clearly distinguish single units from multi-units. This is described in the Materials and methods, but referred to later in the manuscript.

5) With respect to analyses, there are questions regarding the independence of many of the data points (e.g. neurons recorded simultaneously or from the same animals, same sessions, etc.), and how this interdependence might contribute to some of the findings. Relatedly, could some neural effects can be accounted for by the animals from which they were recorded (and from that particular animal's behavior)?

6) With respect to novelty, some of the ground covered in this report has been covered before. Using a different experimental approach, the study of Knyazeva et al., 2020, Front Neurosci. 14: 306 already suggested that the discharges of many neurons in AC are affected by arousal, that task effects can disappear if effects of arousal have been accounted for, and that there is no systematic difference in response modulation between neurons tuned, or not tuned, to task-relevant sounds. Dissociations of the effects of different non-auditory factors on sound responses in AC have also been described by Zhou et al., 2014, and by Carcea et al., 2017. This work should be discussed.

[Editors' note: further revisions were suggested prior to acceptance, as described below.]

Thank you for resubmitting your article "Dissociation of task engagement and arousal effects in auditory cortex and midbrain" for consideration by *eLife*. Your revised article has been reviewed by two peer reviewers, and the evaluation has been overseen by a Reviewing Editor and Andrew King as the Senior Editor.

A concern remains. One reviewer noted “it appears that data from only about 20 behavioral sessions entered analyses”. The reviewer was concerned that neurons simultaneously recorded in the same session were statistically more dependent than neurons recorded in different sessions. In your revised manuscript please include the number of animals from which the pupil data and the neuronal data were obtained, state why one animal was not considered for neuronal data (it appears that there were four animals in total of which three provided neuronal data) and include the total number of sessions for pupil data and the total number of sessions for neuronal data in each animal. The reviewer was concerned about what can be inferred from these sample sizes. Please also include a statement that the results on the effects of task engagement may not apply to all auditory tasks.

---

## [Author Response]

Essential revisions1) The major concern is the use of a continuous readout of arousal (i.e. pupil diameter) with a binary readout of task-engagement (i.e. the block the animal is in at any moment). The reviewers would like to know if task engagement be explained more rigorously as a continuous rather than binary variable. One notes that when training and testing animals on appetitive behaviors, task engagement can wax and wane within a single block, across an experimental recording session, or across days of behavioral testing. Such changes in engagement can be inferred, for example, as strings of (seemingly) easy trials in which the animal does not answer correctly. The authors should attempt to quantify through behavioral analysis (running lapse rate, lick latency, etc) whether and how task engagement may be changing within and across task blocks. Alternatively, the authors could clearly explain that their binary encoding of engagement has limitations and may not actually describe the animal's engagement at any given moment.

We agree completely that treating task engagement as a binary variable is problematic. In fact, this concern was a key motivation for the current study, which determined how much of a canonical “task engagement” effect (reported in many studies—Downer et al., 2015; Fritz et al., 2005, 2003; Knudsen and Gentner, 2013; Kuchibhotla et al., 2017; Lee and Middlebrooks, 2011; Otazu et al., 2009; Ryan and Miller, 1977; Yin et al., 2014) can be explained by pupil-indexed arousal. A change in pupil size may itself also not reflect a single neural process, but we argue in the Discussion that it is closer constituent of a behavioral state than the binary task engagement variable.

More to the reviewer’s point, the effects of task engagement that cannot be explained by pupil size are still likely not to reflect a single, binary variable. Early in the project, we explored several more fine-grained models that broke down engagement based on behavioral performance, but these were not pursued because they did not provide additional explanatory power in our initial dataset. However, we revisited this question for the full dataset and identified an effect of task performance that was not apparent earlier.

In the revised manuscript, we compared the magnitude of task engagement effects on neural activity to behavioral performance (measured by sensitivity, *d*’) on each behavioral block. *d*’ is higher when animals tend to respond correctly (high hit rate, low false alarm rate). When we tested the relationship between *d*’ and engagement effects, we found a correlation in A1 but not IC. We also found no relationship between behavioral performance and pupil-related changes in neural activity in either area.

To study performance-dependent variability on a finer timescale, we formulated a more complex model with two additional regressors—the moving average of hit rate and false alarm rate over a 3 minute window (~25 trials). A model including these new regressors explained some additional neural variability over the original binary engagement model. However, it did not improve accuracy over the block-wise performance-dependent model described in the previous paragraph. We elected to not include this model in the revision because of its complexity and the fact that the block-wise performance model demonstrated the critical result that some task-related effects depend on behavioral performance.

In sum, our new analysis confirms that, even after accounting for pupil-related effects on neural activity, task engagement is in fact not a binary variable. This result also provides additional evidence for functional differences between A1 and IC. We now include the analysis of performance dependence in a new Figure (Figure 4):

These results are also described in the text:

“The analysis above treated task engagement as a discrete, binary variable, switching between active and passive states. However, behavioral performance, measured by d’, varied substantially between animals and behavior blocks (Figure 1C). We reasoned that performance represents a more graded measure of task engagement and thus may explain variability in task-related changes across experiments. Indeed, we found a significant correlation between d’ and the unique variance explained by task in A1 (*r*^2^ task unique; *r* = 0.303, *p* = 0.00019; Figure 4A, left). The same relationship was not observed in IC (*r* = 0.069, *p* = 0.580; Figure 4A, right), nor was there a relationship between d’ and unique variance explained by pupil in either area (*r*^2^ pupil unique; A1: *r* = 0.113, *p* = 0.139; IC: r = -0.017, *p* = 0.892, Figure 4B). Thus, in A1, effects of task engagement cannot be described by a binary variable, even after dissociating effects of pupil. Instead, effects of task engagement are comprised of at least one process that varies continuously between more and less accurate behavior.”

To emphasize that finer timescale analysis such as the moving average model might be valuable when adequate data are available, we highlight this as a valuable future direction for identifying important task-related state variables:

“Larger datasets that track continuous fluctuations in performance are likely to be able to measure fluctuations of task engagement effects within behavior blocks.”

2) Can a continuous readout of task engagement better explain neural activity? For many neurons, task-engagement does not provide unique predictive information, yet for others it does (e.g. Figure 3C). If task engagement can be modeled as a continuous rather than binary variable, is it still true that "some apparent effects of task engagement should in fact be attributed to fluctuations in arousal" (Abstract)? There is a concern that the current analysis maybe a floor on task-related modulations since it assumes constant engagement throughout a task block.

We agree that “task engagement” is not a discrete, binary variable. Our formulation was a simplification, motivated by the limited statistical power available in the experimental data. However, the analysis described above demonstrates an additional dependence on task performance that we have added in a new figure in the revised manuscript. See reply to #1 for details.

We would also like to point out that theories have been developed around network attractor states, and that engaging in a task could actually produce a stable shift in neural network dynamics (Le Camera et al. 2019). Thus, while unlikely to be the whole story, the idea of a discrete (if not binary) variable is not completely far-fetched. We have also clarified issues around the meaning of “task engagement” in the Discussion.

“It remains unclear how smoothly internal state can vary. Theories of network attractors suggest that there may in fact be discrete changes in brain state associated with different behavioral context (La Camera et al., 2019). Thus, while clearly task engagement is not binary, it could still comprise multiple metastable states.”

3) Can neural heterogeneity be attributed to animal-to-animal behavioral variability? Even if task engagement does not vary within a task block for any one animal, it may indeed vary across animals. In theory, the actual task engagement of some animals might more closely mirror the block design that the experimenters are imposing, and some animals may simply have a higher level of engagement than others. This could mean that some results that are currently attributed to population-level heterogeneity (e.g. some A1 neurons do this, while others do that) might actually be attributed to animal-to-animal heterogeneity as opposed to distinct neural populations. For example, the authors state that for a subset of neurons, persistent task-like activity after a block change can be accounted for by pupil size, whereas for other neurons this effect cannot (Figure 7). The authors should confirm that key findings are consistent across animals and not related to degrees of task engagement (see point #1). If the findings are not consistent across animals but can be explained by each animals' unique behavior, this would also be really cool.

Thanks for raising this point. When we went back to the data to look at the relationship between performance and task effects, we also observed differences in both the size of task engagement effects in A1 and behavioral performance (*d*’) between animals. These were correlated, and when we performed multiple regression, we found that the differences in neural activity between animals could be explained completely by *d*’. So indeed, as the reviewer suggests, there are differences between animals, which are reflected in *d*’. These analyses are detailed in new figures and in the Results (Figure 4):

“We also observed that both *d*’ and task engagement effects in A1 differed between animals (Figures 4, Figure 4—figure supplement 1). We wondered if the differences in neural modulation could be fully explained by behavioral performance or if they reflected additional between-animal differences. To answer this question, we performed a multiple regression with both *d*’ and animal identity as independent variables. This analysis revealed that in A1 *d*’ could fully explain the differences in modulation for task engagement (*d*’: *F* = 16.0, *p* = 0.000093; animal: *F* = 0.66, *p* = 0.52). Neither *d*’ or animal significantly predicted task engagement effects in IC (*d*’: *F* = 1.11, *p* = 0.29; animal: *F* = 0.22, *p* = 0.80). A similar analysis of pupil-related effects revealed that *r*^2^ pupil unique did not depend on *d*’, although it did depend on the amount of pupil variability within an experiment (Figure 4—figure supplement 1). Thus, differences in task engagement effects between animals could be explained by differences in the accuracy with which they performed the task.”

As suggested, we also compared changes in post- versus pre-passive responses across animals and found no significant difference. This is reported in the revision:

“This change was not significantly different between animals in A1 (*F* = 0.669, *p* = 0.516, one-way ANOVA) or in IC (*F* = 0.446, *p* = 0.643).”

4) With respect to recordings, the authors should clearly distinguish single units from multi-units. This is described in the Materials and methods, but referred to later in the manuscript.

This is a good question. We compared multi- and single-unit data and didn’t find significant differences between arousal or engagement effects in either A1 or IC. We include this analysis in a new supplemental figure.

“These effects were not significantly different between single- and multi-unit data in either area (Figure 3—figure supplement 1).”

5) With respect to analyses, there are questions regarding the independence of many of the data points (e.g. neurons recorded simultaneously or from the same animals, same sessions, etc.), and how this interdependence might contribute to some of the findings. Relatedly, could some neural effects can be accounted for by the animals from which they were recorded (and from that particular animal's behavior)?

We agree, it is important to consider potential differences between animals. Our new analysis finds that there are in fact differences in behavioral performance (*d*’) between animals. While there are no significant differences in task engagement effects between animals, there is a relationship between performance (*d*’) and neural effects (see reply to #1 and #3 above). We have added figures describing these results (Figures 4, Figure 4—figure supplement 1). See quoted figure/text in the reply to #1 above.

6) With respect to novelty, some of the ground covered in this report has been covered before. Using a different experimental approach, the study of Knyazeva et al., 2020 already suggested that the discharges of many neurons in AC are affected by arousal, that task effects can disappear if effects of arousal have been accounted for, and that there is no systematic difference in response modulation between neurons tuned, or not tuned, to task-relevant sounds. Dissociations of the effects of different non-auditory factors on sound responses in AC have also been described by Zhou et al., 2014, and by Carcea et al., 2017. This work should be discussed.

The studies cited by the reviewer are quite relevant to the current study, in having demonstrated effects of behavioral state that are distinct from task engagement. In the case of Zhou et al., the motor activation is likely to be closely related to pupil-indexed arousal measured in our study. In the case of Knyazeva et al. and Carea et al., it is clear that changing the rules of behavior impacts neural coding and further amplifies the points that “task engagement” is not a binary variable. While none of these studies measured pupil, it is likely that some of the changes they report could be explained by pupil-indexed arousal while others may be explained by mechanisms that produce performance-dependent changes during task engagement.

In the spirit of the task structure effects (Knyazeva and Carea studies), we tested for a dependence on task difficulty in the original manuscript (now Figure 6—figure supplement 2). We now cite these previous studies in our description of this analysis in the Results, as well as incorporate all of this literature into more general points in the Introduction and Discussion:

“Introduction: Task-related changes in the activity of auditory neurons have been attributed to temporal expectation (Jaramillo and Zador, 2011), reward associations (Beaton and Miller, 1975; David et al., 2012), self-generated sound (Eliades and Wang, 2008), and non-sound related variables, such as motor planning (Bizley et al., 2013; Huang et al., 2019), motor activity (Schneider et al., 2014; Zhou et al., 2014), degree of engagement (Carcea et al., 2017; Knyazeva et al., 2020), and behavioral choice (Tsunada et al., 2015).”

“Results: Task engagement effects could also depend on the difficulty or other structural elements of the task (Carcea et al., 2017; Knyazeva et al., 2020). We considered whether state-dependent effects varied with task difficulty but found no differences between pure tone, high SNR tone-in-noise (easy), and low SNR tone-in-noise (hard) conditions (Figure 6—figure supplement 2).”

“Discussion: The current results illustrate that a binary task engagement variable is better described by a combination of pupil size and accuracy of behavioral performance. Other characterizations of task engagement have argued similarly that the degree of engagement can vary, even when relevant acoustic stimuli are held fixed (Atiani et al., 2009; Carcea et al., 2017; Knyazeva et al., 2020; McGinley et al., 2015; Zhou et al., 2014). Larger datasets that track continuous fluctuations in performance are likely to be able to measure fluctuations of task engagement effects within behavior blocks. It remains unclear how smoothly internal state can vary. Theories of network attractors suggest that there may in fact be discrete changes in brain state associated with different behavioral contexts (La Camera et al., 2019). Thus, while task engagement is clearly not binary, it could still comprise multiple metastable states.”

[Editors' note: further revisions were suggested prior to acceptance, as described below.]

A concern remains. One reviewer noted “it appears that data from only about 20 behavioral sessions entered analyses”. The reviewer was concerned that neurons simultaneously recorded in the same session were statistically more dependent than neurons recorded in different sessions. In your revised manuscript please include the number of animals from which the pupil data and the neuronal data were obtained, state why one animal was not considered for neuronal data (it appears that there were four animals in total of which three provided neuronal data) and include the total number of sessions for pupil data and the total number of sessions for neuronal data in each animal.

Thanks for bringing these questions to light. Some of this information was implicit in the text, but we agree, it is much clearer to report it explicitly, as suggested. A total of four animals were included in the study (as shown in Figure 1). Neurophysiological data was collected from both A1 and IC in two animals (animals B and R) and only from A1 or IC in the remaining two (animals T and L, respectively). We have added information about animal and site count to the Results:

Pupil data sessions: “Mean pupil size (measured by the major axis diameter) was consistently larger during the active condition (n=35 sites total, animal L: n=13, B: n=5, R: n=16, T: n=1, p = 0.0133, hierarchical bootstrap, Figure 1D).”

Neural data sessions: “We recorded extracellular spiking activity from the primary auditory cortex (A1, n = 129 units total, animal B: 6 sites/88 units, R: 1 site/13 units, T: 1 site/28 units) and the inferior colliculus (IC, n = 66 units total, animal B: 4 sites/7 units, L: 13 sites/46 units, R: 10 sites/13 units) of ferrets, while they switched between active engagement and passive listening.”

The reviewer was concerned about what can be inferred from these sample sizes.

General observations in the current data alleviated concerns about the main results, namely the large variability of effects within a single array recording (e.g., Figure 4), exemplified by the examples in Figure 2, where two neurons recorded simultaneously show very different effects of behavioral state.

However, this comment did motivate us to investigate a more rigorous statistical approach. We implemented a hierarchical bootstrap analysis developed explicitly to control for the type of bias that can arise in array recordings (Saravanan et al., 2020 *arXiv*), and we used it to assess significance of differences across populations in different experimental conditions. Because of this, reports of statistical tests have been updated in several places. Note that the hierarchical bootstrap method allows us to directly calculate the probability of the tested hypothesis being true. With 10000 bootstrap resamples, the minimum possible probability (or p-value) is 10^-5^, and therefore we report “p<10^-5^” when zero bootstrap samples provide evidence supporting the null hypothesis.

We have revised the presentation of statistical tests to highlight the new approach:

“Several analyses assessed changes in the median of repeatedly sampled data, *e.g.*, average pupil size measured across multiple behavior blocks (Figure 1) or average state-dependent modulation measured across multiple units (Figures 3, 5-8). In this case, significant differences were assessed using a hierarchical bootstrap test (Saravanan et al., 2020). The hierarchical bootstrap is nonparametric, like the more traditional Wilcoxon signed-rank test. However, a traditional test that treats each unit as an independent measure could potentially be biased if a single array recording produced spuriously large effects. The hierarchical bootstrap controls for this possibility by resampling first across recording sessions and then within each of those sessions. The bootstrap analysis was run for 10,000 iterations, so that the minimum measurable p-value was 10^-5^. Results that returned *p*=0 after this many iterations are reported as *p* < 10^-5^.”

We made numerous small changes to report the new test. Rather than detailing all the changes, following is a list of analyses where a Wilcoxon signed-rank test or permutation test (for correlation coefficient) was replaced by a hierarchical bootstrap. In these cases, the basic result (*p*<0.05 or p>0.05) was unchanged:

There were two instances where the hierarchical bootstrap test produced a different result than the previous version of the manuscript. Neither of these analyses is critical to the main findings in the study:

Figure 5: In IC, the average change in spike rate uniquely attributable to task engagement (*MI_AP_* task-unique) was previously reported to be significantly less than zero but now tests at p = 0.062. We have revised the text: “In IC, average *MIAP task-only* was not different from zero (*MIAP task-only* median: -0.010, *p* = 0.310), but *MI_AP_ task-unique* showed a trend toward being negative (17/49 positive, median: -0.037; *p* = 0.062).”

We did find that accounting for effects of pupil consistently decreased *MI_AP_* (i.e., *MI_AP_* task unique < *MI_AP_* task-only) in both A1 and IC. We report that in the legend for Figure 5: “*MI_AP_* shifted to more negative values on average (A1 median *MI_AP_ task-only* = 0.069, *MI_AP_ task-unique* = 0.027, *p* =0.0005; IC: median *MI_AP_ task-only* = -0.010, *MI_AP_ task-unique* = -0.037, *p* = 0.049, hierarchical bootstrap).”

Figure 6—figure supplement 1: The comparison between *MI_AP_* task-unique for on- versus off-BF units in A1 was previously reported as significant, but now we find p=0.221. We have revised the text: “In A1, there was no difference in median *MI_AP_* task-unique between modulated on- and off-BF units (on-BF: median 0.082, n = 28; off-BF: -0.012, n = 24; p = 0.221, hierarchical bootstrap) or unmodulated units (on-BF: -0.002, n = 44; off-BF: -0.019, n = 77; p = 0.136).”

Please also include a statement that the results on the effects of task engagement may not apply to all auditory tasks.

As suggested, we have clarified this point in the Discussion:

“The effects of task engagement reported here are specific to the tone detection task and are likely to differ if task structure is changed. In contrast, effects of pupil modulate activity throughout active and passive states, and are likely to be similar in different task conditions (Schwartz et al., 2020). “